# Chiral cell sliding drives left-right asymmetric organ twisting

**Mikiko Inaki[1], Ryo Hatori[1], Naotaka Nakazawa[1], Takashi Okumura[1], Tomoki Ishibashi[1], Junichi Kikuta[2], Masaru Ishii[2], Kenji Matsuno[1†]\*, Hisao Honda[3†]\***

[1]Department of Biological Sciences, Graduate School of Science, Osaka University, Toyonaka, Japan; [2]Department of Immunology and Cell Biology, Graduate School of Medicine, Osaka University, Suita, Japan; [3]Department of Physiology and Cell Biology, Graduate School of Medicine, Kobe University, Kobe, Japan

**Abstract** Polarized epithelial morphogenesis is an essential process in animal development. While this process is mostly attributed to directional cell intercalation, it can also be induced by other mechanisms. Using live-imaging analysis and a three-dimensional vertex model, we identified 'cell sliding,' a novel mechanism driving epithelial morphogenesis, in which cells directionally change their position relative to their subjacent (posterior) neighbors by sliding in one direction. In *Drosophila* embryonic hindgut, an initial left-right (LR) asymmetry of the cell shape (cell chirality in three dimensions), which occurs intrinsically before tissue deformation, is converted through LR asymmetric cell sliding into a directional axial twisting of the epithelial tube. In a *Drosophila* inversion mutant showing inverted cell chirality and hindgut rotation, cell sliding occurs in the opposite direction to that in wild-type. Unlike directional cell intercalation, cell sliding does not require junctional remodeling. Cell sliding may also be involved in other cases of LR-polarized epithelial morphogenesis.

DOI: https://doi.org/10.7554/eLife.32506.001

**\*For correspondence:**
kmatsuno@bio.sci.osaka-u.ac.jp
(KM);
hihonda@hyogo-dai.ac.jp (HH)

[†]These authors contributed equally to this work

**Competing interests:** The authors declare that no competing interests exist.

## Introduction

Left-right (LR) asymmetry is a basic feature of metazoan development. Although the outer structures of bilaterians appear bilaterally symmetric, their internal organs are often LR asymmetric in morphology and positioning. The mechanisms for LR asymmetry formation have been studied extensively in vertebrates (*Hirokawa et al., 2012*; *Nakamura and Hamada, 2012*; *Vandenberg and Levin, 2013*; *Yoshiba and Hamada, 2014*). In mice, a leftward flow of extraembryonic fluid, called nodal flow, is generated in the node by the clockwise rotation of cilia. The nodal flow breaks the LR symmetry and eventually induces left-side-specific gene expressions, including *Nodal*, *Lefty*, and *Pitx-2*, the so-called nodal cassette genes (*Nonaka et al., 2002*; *Nonaka et al., 1998*; *Okada and Hirokawa, 1999*). In addition to nodal flow, various other mechanisms are used in different vertebrates, but they also ultimately lead to the left-side-specific expressions of nodal cassette genes (*Levin, 2005*), which influence the LR asymmetric morphology and positioning of internal organs (*Huang et al., 2014*; *Yoshiba and Hamada, 2014*). However, it was recently reported that the LR asymmetric looping of the zebrafish heart develops by a tissue-intrinsic mechanism, independent of *nodal* (*Noël et al., 2013*). Therefore, parallel mechanisms are involved in the LR asymmetric development of vertebrates.

LR asymmetry has been reported at the cellular level, as well as in organs (*Chen et al., 2012*; *Wan et al., 2011*; *Xu et al., 2007*). Many mammalian cell lines adopt an LR asymmetric shape when cultured on a micropattern (*Chen et al., 2012*; *Raymond et al., 2016*; *Wan et al., 2011*; *Worley et al., 2015*). The LR asymmetric cell shape is termed 'cell chirality' because the cell shape

**eLife digest** Many organs arise from simple sheets and tubes of cells. During development these sheets bend and deform into the more complex shape of the final organ. This can be seen, for example, in the hindgut of fruit flies, which is an organ that is equivalent to our intestines. Initially, the hindgut is a simple tube of cells. Later the hindgut develops a twist to the left that renders its right and left sides non-symmetrical. During twisting, the cells in the hindgut also change shape. It was not known how this shape change and other behaviors of the cells cause the hindgut to twist.

Inaki et al. have now filmed how the hindgut develops in live fruit flies and produced computer simulations of the development process. The results suggest that a previously unidentified type of cell behavior called 'cell sliding' is responsible for twisting the hindgut. During sliding, the cells stay in contact with their neighbors as they move in a single direction. Sliding is triggered by the cells in the hindgut taking on a more symmetrical shape.

Cell sliding may prove to be a common way to shape organs, many of which feature non-symmetrical twisted tubes of cells. In the future, learning how to control cell sliding could help researchers to create organs and biological structures in the laboratory that could be used in organ transplants and regenerative medicine.

DOI: https://doi.org/10.7554/eLife.32506.002

cannot be superimposed on its mirror image. Cell chirality is observed in both the shape and behavior of cells. Cultured zebrafish melanophores show chirality in cellular locomotion and in cytoplasm swirling (*Yamanaka and Kondo, 2015*). Fibroblasts from human foreskin seeded on a micropattern exhibit a chiral swirling of actin fibers (*Tee et al., 2015*), and cultured neutrophils show LR-biased movement in the absence of positional cues (*Xu et al., 2007*). However, the physiological roles of cell chirality in vertebrates remain unknown.

An in vivo function of cell chirality was first discovered in the *Drosophila* embryonic hindgut (*Taniguchi et al., 2011*), which first forms as a bilaterally symmetric structure and then rotates 90° counterclockwise as viewed from the posterior, showing dextral looping (*Hozumi et al., 2006*). The posterior end of the hindgut does not rotate, and thus the hindgut twists as a whole. The hindgut epithelial cells are probably responsible for this rotation, since the LR defect in hindgut rotation in mutants is fully rescued when the responsible genes are expressed specifically in hindgut epithelial cells (*Hozumi et al., 2006*; *Taniguchi et al., 2011*). Before the directional rotation begins, the anterior-posterior axis of the hindgut can be defined, because its simple tubular structure extends in the anterior-posterior direction, and the hindgut epithelial cells exhibit an LR asymmetric shape of their apical surface with respect to the anterior-posterior axis (*Taniguchi et al., 2011*). Because hindgut epithelial cells have apical-basal polarity, like other epithelial cells, their LR asymmetric shape can be regarded as chiral. The LR asymmetric shape eventually disappears and the cells become symmetric after the rotation (*Taniguchi et al., 2011*). A previous computer simulation showed that the introduction and subsequent dissolution of cell chirality are sufficient to induce the rotation of a model epithelial tube (*Taniguchi et al., 2011*). During the rotation, neither cell proliferation nor cell death occurs in the hindgut (*Lengyel and Iwaki, 2002*; *Wells et al., 2013*), indicating that cell-shape changes and/or cell rearrangements are involved in this process. Together, these observations indicate that cell chirality drives the counterclockwise rotation of the hindgut. However, the cellular dynamic mechanism by which cellular chirality is converted into axial rotation of the hindgut remains unknown.

In addition to cell chirality, various other cellular dynamic mechanisms contribute to the morphological changes of epithelial tissues, such as cell intercalation and cell deformation. Cell intercalation involves anisotropic cell-boundary remodeling (*Bertet et al., 2004*). For example, if cells intercalate in a medial direction, the tissue becomes narrower and elongates along the axis perpendicular to the medial direction (*Honda et al., 2008*; *Tada and Heisenberg, 2012*; *Uriu et al., 2014*). Polarized cell intercalation is important in convergent extension, which induces morphological changes in early embryogenesis, such as the germband extension in *Drosophila* and the dorsal mesoderm extension in zebrafish and *Xenopus* (*Bertet et al., 2004*; *Shih and Keller, 1992*). Convergent extension is also required for organogenesis. For example, tubular structures, such as the *Drosophila* trachea and

hindgut and the vertebrate kidney and cochlea, elongate by convergent extension (*Chen et al., 1998*; *Iwaki and Lengyel, 2002*; *Karner et al., 2009*; *Wang et al., 2005*). Cell intercalation also contributes to LR asymmetric morphogenesis. For example, LR biased junctional remodeling induces the directional rotation of the *Drosophila* male genitalia (*Sato et al., 2015a*). Cell deformation is another mechanism that plays important roles in epithelial morphogenesis. During gastrulation and neurulation, the apical constriction of epithelial cells is important for invagination and tubular structure formation (*Inoue et al., 2016*; *Munjal and Lecuit, 2014*). Thus, one of these cellular dynamic mechanisms or an as-yet undescribed mechanism might be involved in the cell-chirality-driven hindgut rotation.

Here, to investigate the cell dynamics underlying the counterclockwise rotation of the *Drosophila* hindgut, we first performed live imaging and revealed a novel cellular behavior we call 'chiral cell sliding,' in which cells directionally change their position relative to their subjacent (posterior) neighbors by sliding in one direction. We found that this cell sliding did not require junctional remodeling. We then performed an improved 3D computer simulation analysis of the cell-chirality-driven hindgut rotation, which confirmed that chiral cell sliding is a dynamic cellular process induced by the dissolution of cell chirality. Our findings collectively showed that chiral cell sliding is a novel mechanism that induces anisotropic changes in tissue morphology.

## Results

### *Drosophila* hindgut epithelial cells show chiral sliding

Our previous genetic analyses suggested that the hindgut itself generates the mechanical force that drives its rotation (*Taniguchi et al., 2011*). Here, we found that an explanted wild-type hindgut rotated counterclockwise in vitro, confirming that the hindgut itself generates an active force to accomplish the rotation (*Figure 1—figure supplement 1*, *Figure 1—video 1*). To investigate what kind of cellular behaviors contribute to the hindgut rotation in vivo, we observed the movement of hindgut epithelial cells by live imaging. We used a Gal4-UAS system (*Brand and Perrimon, 1993*). The expression of UAS-*redstinger* (a nuclear marker) and UAS-*myrGFP* (a membrane marker) was driven by a hindgut-specific driver, *byn*-gal4 (*Barolo et al., 2004*; *Iwaki and Lengyel, 2002*; *Pfeiffer et al., 2012*). We aligned embryos in the anterior-posterior direction, observed them from the dorsal side, and took movies for 2 hr from the onset of hindgut rotation. The hindgut started to rotate at late embryonic stage 12, when the germ band was almost completely retracted (*Figure 1A1*). Before rotation, the hindgut is LR symmetric and has a hook-like shape pointing to the ventral side of the embryo (*Lengyel and Iwaki, 2002*). The hindgut twisted 90° counterclockwise, and this rotation was fully complete by 2 hr (*Figures 1A1–3*, *Figure 1—video 2*). After completion of the rotation, the hindgut had a hook shape that pointed rightward (*Figure 1A3*).

We analyzed the cell displacement by observing the positions of nuclei. This analysis was done in 2D, because the z resolution (1.5–1.7 µm) was much lower than the xy resolution (0.313 µm). To minimize the loss of displacement by projecting the image to 2D, we analyzed only the central three columns of cells (*Figures 1B,C1–3*). It is important to consider the hook shape of the hindgut when analyzing the displacement. Cells located in the hook part of the anterior region of the hindgut are affected more by the rotation of the whole hindgut than cells located in the posterior root part. Even when cells do not change their relative positions, they appear to move leftward if they are located posterior to the hook peak, whereas they appear to move rightward when located anterior to the hook peak (*Figure 1B*). To minimize this apparent LR asymmetric displacement that is intrinsically associated with the hook-like structure, we analyzed the cell displacement in the sub-root part of the hindgut, which is about 20–40% of the hindgut in length (red and blue boxes in *Figure 1B*).

Using time-laps movies, we tracked cells located in the dorsal side of the hindgut by their nuclear position in individual time frames (5 min intervals) (red boxes in *Figure 1B*). The cells changed their position relative to the cells located below them (posterior in the embryo) by sliding leftward (*Figures 1C1–3*, *Figure 1—video 3*). We next measured the relative displacement in the x direction of each cell against its subjacent neighbor [placed at (0,0) in xy coordinates] every 30 min (*Figure 1E*). 'Minus' displacement indicates leftward movement, while 'plus' displacement indicates rightward movement. The displacement of wild-type epithelial cells in the dorsal side was about −0.5 µm, indicating that the cells moved significantly leftward in relation to their subjacent

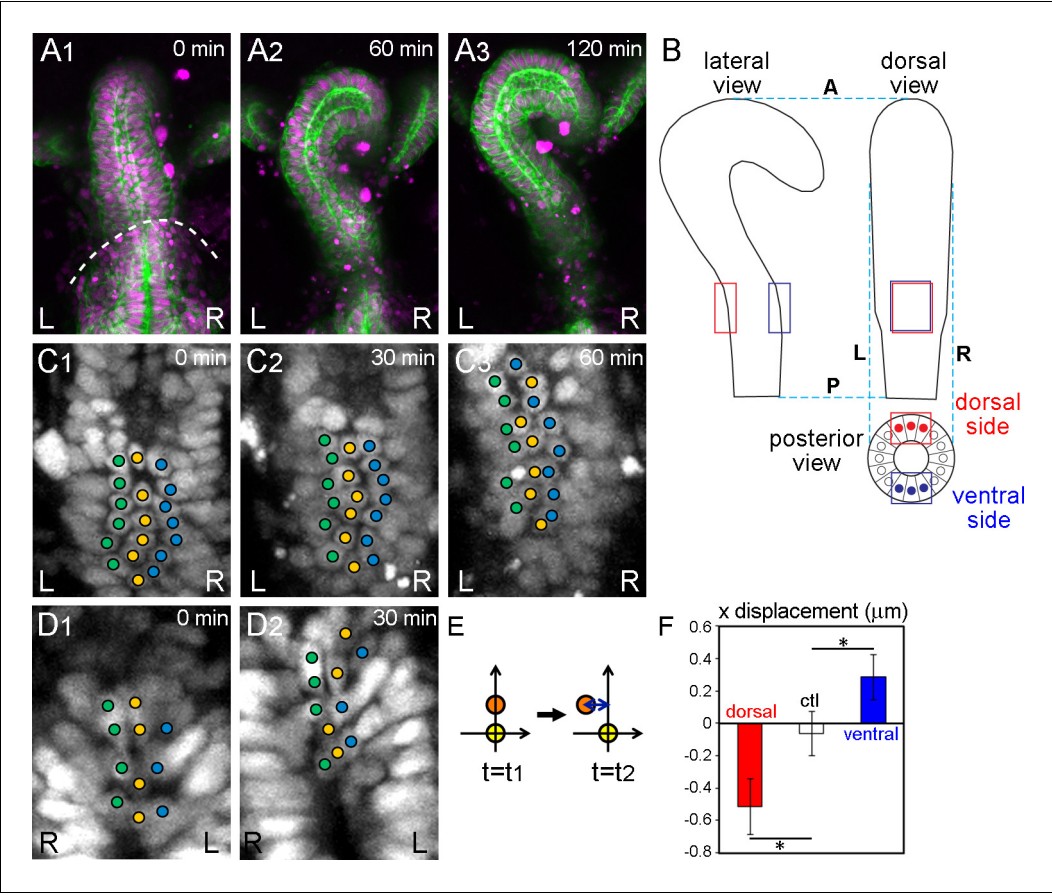

**Figure 1.** Wild-type hindgut epithelial cells slide leftward. (**A**) *Drosophila* embryonic hindgut visualized by UAS-*myr-GFP* (membrane in green) and UAS-*redstinger* (nuclei in magenta) driven by *byn*-Gal4, starts to rotate at late stage 12, when the germ band is almost completely retracted (dashed line in A1). The hindgut rotates anticlockwise (**A2**) and completes the rotation in 2 hr, exhibiting a rightward pointing hook shape (**A3**). Time after the start of rotation is indicated at upper right. (**B**) Schematic of the hindgut before rotation. Lateral (left), dorsal (right), and posterior (bottom) views are shown. The hindgut is LR symmetric with the hook shape pointing in the ventral direction. The regions observed in the displacement measurements are indicated by red (dorsal) and blue (ventral) rectangles. (**C, D**) Cell nuclei in the dorsal (**C**) and ventral (**D**) sides of the hindgut, schematically shown as red and blue circles in B (posterior view), respectively, visualized by *byn*-Gal4 and UAS-*redstinger* (**C**) or UAS-*stinger* (**D**). Three central columns of cells are marked by colored circles. Cells slid leftward on both the dorsal and ventral sides. (**E**) Schematic of the displacement quantification. The coordinates of two cells located along the anterior-posterior axis at two time points were observed. The subjacent cell was set at (0,0), and the relative displacement of the upper cell in the x direction was measured. (**F**) Quantification of the cell movement during wild-type hindgut rotation. Dorsal cells showed minus directional movement (red bar, n = 126, N = 5) while ventral cells showed plus directional movement (blue bar, n = 78, N = 12), indicating that the cells on both sides moved leftward. Cells in the rectum region (shown as a control, ctl) did not show LR displacement (open bar, n = 91, N = 5). Error bars indicate SEM. *, p<0.05. For all images, anterior is up. A, anterior; P, posterior; L, left; R, right.
DOI: https://doi.org/10.7554/eLife.32506.003

The following video, source data, and figure supplements are available for figure 1:

**Source data 1.** Source data for *Figure 1F*.
DOI: https://doi.org/10.7554/eLife.32506.006
**Figure supplement 1.** Explanted wild-type hindgut rotates counterclockwise in vitro.
DOI: https://doi.org/10.7554/eLife.32506.004
**Figure supplement 2.** Cell movement in the root-most part of wild-type hindgut.
DOI: https://doi.org/10.7554/eLife.32506.005
**Figure 1—video 1.** Explanted wild-type hindgut rotates counterclockwisein vitro.
DOI: https://doi.org/10.7554/eLife.32506.007

*Figure 1 continued on next page*

*Figure 1 continued*

**Figure 1—video 2.** *Drosophila* embryonic hindgut visualized by UAS-*myr-GFP* (membrane in green) and UAS-redstinger (nuclei in magenta) driven by *byn*-Gal4, rotates counterclockwise as viewed from the posterior.
DOI: https://doi.org/10.7554/eLife.32506.008

**Figure 1—video 3.** Wild-type hindgut epithelial cell movement in the dorsal side.
DOI: https://doi.org/10.7554/eLife.32506.009

**Figure 1—video 4.** Wild-type hindgut epithelial cell movement in the ventral side.
DOI: https://doi.org/10.7554/eLife.32506.010

neighbors (*Figure 1F*, dorsal). As a control, we measured the LR displacement in the root-most part of the hindgut, where our time-lapse analysis did not detect displacement (*Figure 1—figure supplement 2*) and found that the value was negligible (*Figure 1F*, ctl).

To confirm that this movement was not caused by an apparent displacement caused by the hook-like shape of the hindgut, we also measured cell movement on the ventral side (blue boxes in *Figure 1B*). If cell displacement contributed to the rotation, the cell displacement in the ventral side should occur in the opposite direction to that in the dorsal side (*Figure 1B*). In contrast, an apparent displacement associated with the hook-like structure of the hindgut should be observed in the same direction, even on the ventral side. For this experiment, we used two-photon confocal microscopy, because the ventral side is more than 50 µm deep from the dorsal surface of the embryo. We visualized the cell nuclei using UAS-*stinger* (a nuclear GFP) (*Barolo et al., 2004*) or UAS-*NLS-tdTomato* with *byn*-Gal4, by the same technique used for the dorsal side (*Figures 1D1* and *2*, *Figure 1—video 4*). Although we could only obtain data from low numbers of cells because most images of the deep ventral region gave poor signals, even using two-photon microscopy, the results revealed a significant opposite directional movement of cells on the ventral side (*Figure 1F*, ventral). These results together suggested that during the counterclockwise rotation of the hindgut, cells change their position relative to their subjacent cells (posterior neighbors) by sliding in the direction of rotation (*Figure 1C,D,F*). Here, we refer to this novel cellular behavior as 'cell sliding.'

## Cells slide in the opposite direction during inverted hindgut rotation in *Myo31DF* mutants

To investigate the contribution of cell sliding to the directional rotation of the hindgut, we also examined the dynamic hindgut cell behavior in a null mutant of the *Myosin31DF* (*Myo31DF*) gene, which encodes the *Drosophila* ortholog of Myosin ID (*Hozumi et al., 2006*; *Spéder et al., 2006*). In *Myo31DF* mutants, the LR asymmetry of various organs is reversed (*Hozumi et al., 2006*; *Spéder et al., 2006*). More than 80% of the *Myo31DF* embryos showed inverted hindgut rotation, with sinistral looping (*Figures 2A1–3*, *Figure 2—video 1*) (*Hozumi et al., 2006*). In *Myo31DF* mutants, the cell chirality of the hindgut epithelial cells before the onset of rotation is also inverted (*Taniguchi et al., 2011*). Furthermore, the chirality of *Myo31DF* cells is determined cell-autonomously (*Hatori et al., 2014*).

Using the same analysis as for wild type, we examined the behavior of hindgut epithelial cells in the dorsal side of the hindgut during its inverted rotation in *Myo31DF* homozygotes. We found that these cells changed their relative position by sliding rightward, which is the opposite direction to that observed in the wild-type cells (*Figures 2B1–3*, *Figure 2—video 2*), and the ventral cells slid to the left side (*Figure 2C1 and*, *Figure 2—video 3*). Quantitative analysis confirmed that the cells on both the dorsal and ventral sides of the *Myo31DF* hindgut slid in a direction opposite to that of wild-type cells (*Figure 2D*, dorsal and ventral, cf. *Figure 1F*). As a control, we again measured the LR displacement in the root-most part of the hindgut where the rotation was negligible, and confirmed that cell sliding was hardly observed (*Figure 2D*, ctl). These results showed that the direction of cell sliding is consistent with the rotation direction of the wild-type and *Myo31DF* hindgut. In addition, the enantiomorphic status of the cell chirality before rotation also coincided with the directionality of the rotation of the wild-type and *Myo31DF* hindgut (*Taniguchi et al., 2011*). Therefore, the initial cell chirality appeared to determine the direction of the cell sliding.

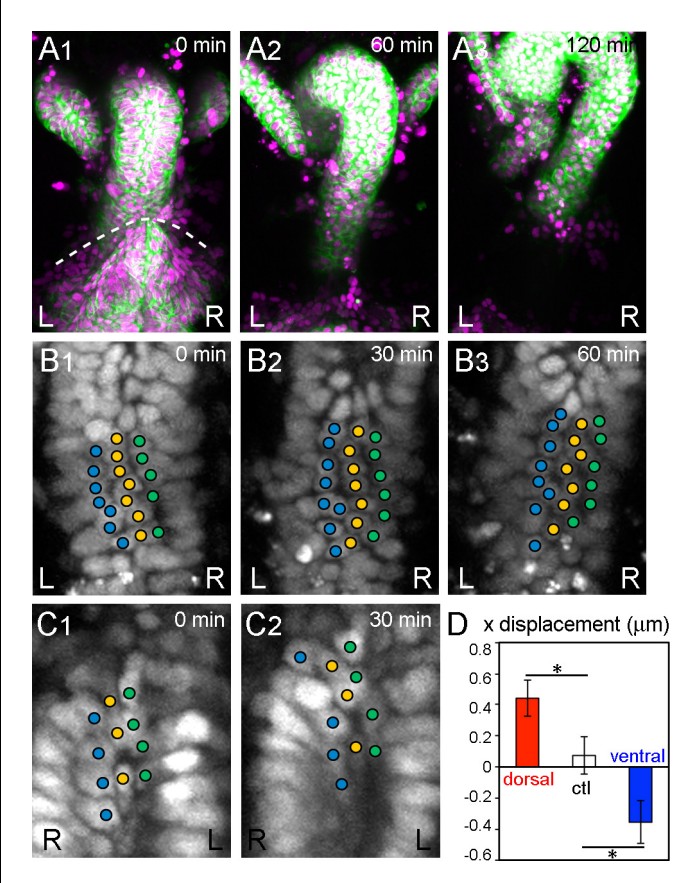

**Figure 2.** During inverted rotation in *Myo31DF* mutants, cells slide in the opposite direction to those in wild type. (**A**) Cell nuclei and membranes of hindgut epithelial cells in *Myo31DF* mutants were visualized as in *Figure 1*. The rotation started at late stage 12, as judged by the position of the germ band (dashed line in A1). *Myo31DF* embryos show inverted rotation of the hindgut (**A2**). After rotation, the hindgut shows a leftward-pointing hook shape (**A3**). Time after the start of rotation is indicated at upper right. (**B, C**) Cell movement on the dorsal (**B**) and ventral (**C**) sides of the hindgut in *Myo31DF* mutants. Cells slide rightward on both the dorsal and ventral sides. (**D**) Quantification of the cell movement in *Myo31DF* mutants as calculated in *Figure 1F*. Dorsal cells showed plus directional movement (red bar, n = 134, N = 5) while ventral cells showed minus directional movement (blue bar, n = 40, N = 6), indicating that the cells on both sides moved rightward, which is opposite to the wild-type cell movement. Cells in the rectum region were used as a control (ctl) (open bar, n = 129, N = 5). Error bars indicate SEM. *, p<0.05. For all images, anterior is up. L, left; R, right.

DOI: https://doi.org/10.7554/eLife.32506.011

The following video and source data are available for figure 2:

**Source data 1.** Source data for *Figure 2D*.
DOI: https://doi.org/10.7554/eLife.32506.012

**Figure 2—video 1.** Myo31DFmutant hindgut visualized by UAS-*myr-GFP* (membrane in green) and UAS-*red-stinger*(nuclei in magenta) driven *bybyn*-Gal4, rotates clockwise as viewed from the posterior.
DOI: https://doi.org/10.7554/eLife.32506.013

**Figure 2—video 2.** Myo31DFmutant hindgut epithelial cell movement in the dorsal side.
DOI: https://doi.org/10.7554/eLife.32506.014

**Figure 2—video 3.** Myo31DFmutant hindgut epithelial cell movement in the ventral side.
DOI: https://doi.org/10.7554/eLife.32506.015

## Cell sliding appears to depend on cell-shape change but not junctional remodeling

It was possible that the cell sliding we observed was coupled with anisotropic cell intercalation, which plays many important roles in epithelial morphogenesis (*Bertet et al., 2004*; *Sato et al.,*

*2015a*). In particular, Sato et al. showed that directional cell intercalation is important in male genitalia rotation, another case of LR asymmetric morphogenesis in *Drosophila* (*Sato et al., 2015a*). To evaluate the potential contribution of LR directional cell intercalation to the cell sliding observed in the hindgut epithelium, we analyzed junctional remodeling, which is a critical feature of cell intercalation (*Bertet et al., 2004*). To investigate the junctional dynamics during cell sliding, we tracked the cell boundaries using UAS-*myrGFP* and *byn*-gal4 in vivo (*Figure 3A,B*). The results revealed that cells maintained their junctions during cell sliding in both the wild type and the *Myo31DF* mutant in most cases (*Figure 3A,B*, *Figure 3—video 1*, *Figure 3—video 2*). The frequency of cell-intercalation events accompanied by junctional remodeling was 9.2 and 6.0% for wild type and the *Myo31DF* mutant, respectively (*Table 1*, *Figure 3—figure supplement 1A,B*). Therefore, more than 90% of the cell junctions were not remodeled during cell sliding, suggesting that the cell sliding does not necessarily require junctional remodeling.

Although the frequency of junctional remodeling was low, it still might have been associated with LR directional cell intercalation. Therefore, we also examined the directionality of the cell intercalation during hindgut rotation. When two cells in a column had initial contact and were subsequently separated by another intervening cell, cell intercalation was said to occur, and the direction from which the intervening cell came was determined (*Figure 3—figure supplement 1A,B*). The frequency of leftward versus rightward intercalations was similar in both the wild-type and *Myo31DF* hindguts (*Table 1*), indicating that the direction of cell intercalation during hindgut rotation does not have an LR bias. We also measured the angles of the cell boundaries undergoing junctional remodeling in the wild-type and *Myo31DF* hindgut as described by Sato et al. (*Sato et al., 2015a*). In both the wild-type and *Myo31DF* hindgut, all of the diminishing cell boundaries had an angle with an absolute value less than 30°, and most of them were within ±10° (*Figure 3—figure supplement 1C, D*). These results further indicated that cell intercalation does not have a major role in the LR directional rotation of the hindgut. Instead, this LR symmetrical cell intercalation contributes to elongation of the hindgut in the anterior-posterior direction through convergent extension (*Iwaki and Lengyel, 2002*). However, it might also amplify the LR asymmetry by increasing the length of the tilted cell columns.

In male genitalia rotation, the distribution of Myosin II (MyoII) to the apical cell boundary is anisotropic, which is responsible for the LR asymmetric junctional remodeling (*Sato et al., 2015a*). Here, we investigated the distribution of MyoII in cell boundaries of the hindgut epithelium. We performed live imaging of MyoII using UAS-*sqhGFP* driven by *byn*-gal4 to visualize the medial and junctional MyoII in the apical regions of hindgut epithelial cells. However, we failed to detect clear anisotropy in the distribution of MyoII (*Figure 3—video 3*). This finding was consistent with our observation that an LR bias in cell interactions was not observed in the hindgut epithelium (*Table 1*, *Figure 3—figure supplement 1*).

Our results suggest that the cell sliding is not caused by cell intercalation but by cell deformation that is probably associated with the dissolution of cell chirality during the hindgut rotation, as reported before (*Taniguchi et al., 2011*). To investigate this possibility, we examined the cell-shape change during the cell sliding by measuring the angle changes in the boundaries between cells aligned in a column every 30 min during hindgut rotation (*Figure 3C,D*). We then classified the changes in these angles as counter-clockwise (ccw), unchanged (uc), and clockwise (cw) (*Figure 3E*). The boundaries between cells aligned in a column tended to tilt in a ccw direction in wild type and in a cw direction in *Myo31DF* mutants during hindgut rotation (*Figure 3F*). These results suggested that cell sliding is accompanied by cell deformation associated with the dissolution of cell chirality.

## Simulation of cell chirality-induced cellular behavior by a Vertex model

To verify that the cell sliding observed in vivo is responsible for the hindgut rotation, we performed a computer simulation analysis. We previously constructed a computer model that suggested that the dissolution of cell chirality in the epithelial cells induces rotation of the hindgut epithelial tube (*Taniguchi et al., 2011*). However, this prototype model was a 2D polygonal model artificially converted to a tube in 3D space, in which the vertices moved in 2D (*Taniguchi et al., 2011*). In addition, the model did not precisely recapitulate the architecture of the hindgut epithelial tube, so it could not be used to verify the cellular dynamics during hindgut rotation. Thus, in the present study, we performed a computer simulation using a 3D cell-based vertex model for tissues (*Honda and Nagai, 2015*; *Honda et al., 2008*; *Honda et al., 2004*; *Nagai and Honda, 2001*).

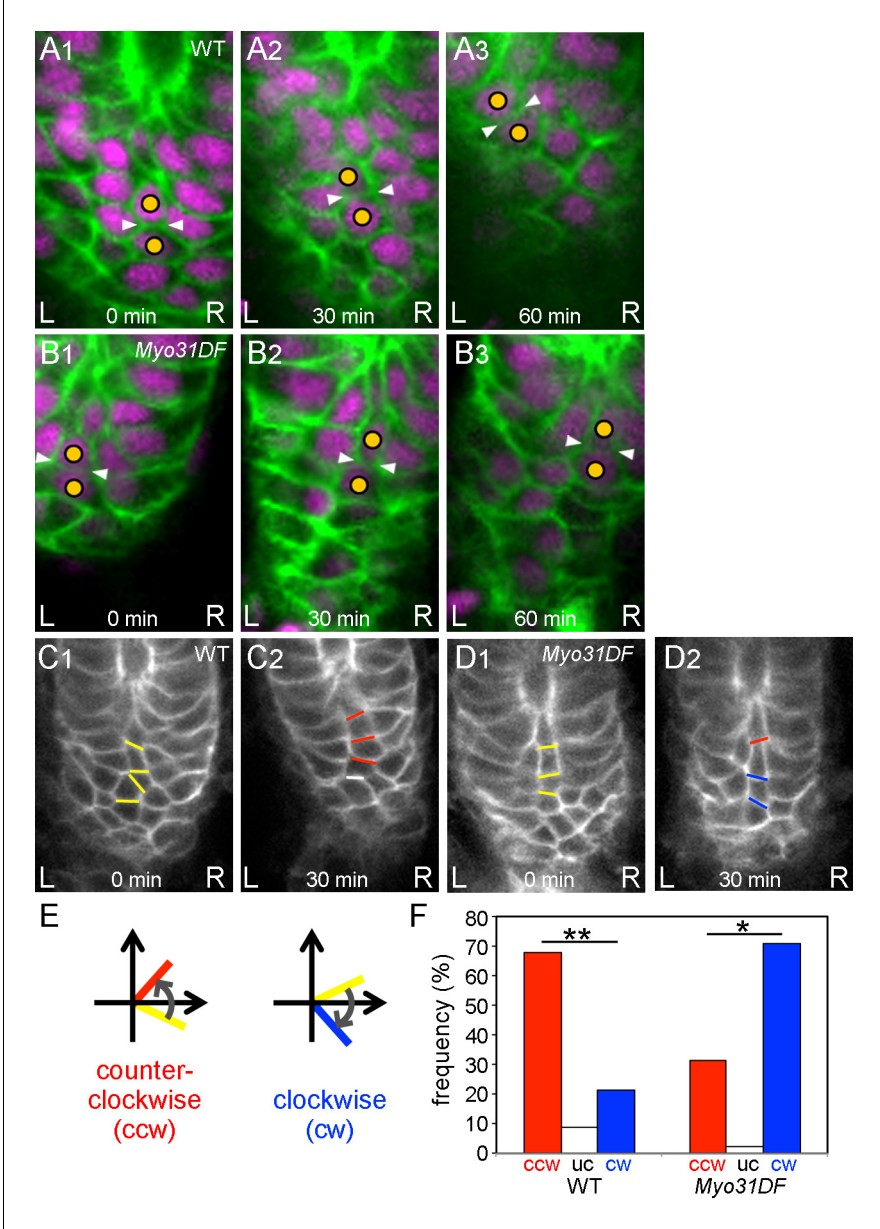

**Figure 3.** Chiral cell-shape change but not junctional remodeling could contribute to directional cell sliding. (**A,B**) Still shots of cell nuclei (magenta) and boundaries (green) of wild-type (**A**) and *Myo31DF* (**B**) hindgut epithelial cells visualized as in *Figure 1A*. Cells maintained their junctions (white arrowheads) during sliding. (**C,D**) Angle change of the cell boundaries during rotation in the wild-type (**C**) and *Myo31DF* (**D**) hindgut. Initial cell boundaries are shown in yellow. Boundaries tilting in a counter-clockwise direction (ccw) are shown in red, while those in a clockwise direction (cw) are in blue. White bars show unchanged (uc) boundaries. (**E**) Schemes for boundary tilting. (**F**) Frequency of boundaries showing ccw, uc, and cw in the wild-type (n = 55, N = 5) and *Myo31DF* hindgut (n = 50, N = 5). **, p<0.01, *, p<0.05. For all images, anterior is up. A, anterior; P, posterior; L, left; R, right.
DOI: https://doi.org/10.7554/eLife.32506.016

The following video, source data, and figure supplement are available for figure 3:

**Source data 1.** Source data for *Figure 3C,D*.
DOI: https://doi.org/10.7554/eLife.32506.018
**Figure supplement 1.** Cell intercalation during LR asymmetric rotation of the hindgut is bidirectional.
DOI: https://doi.org/10.7554/eLife.32506.017
**Figure 3—video 1.** Tracking of two nuclei marked by yellow circles and an in-between boundary of wild-type hindgut epithelial cell.
*Figure 3 continued on next page*

*Figure 3 continued*

DOI: https://doi.org/10.7554/eLife.32506.019

**Figure 3—video 2.** Tracking of two nuclei marked by yellow circles and an in-between boundary of *Myo31DF* mutant hindgut epithelial cell.

DOI: https://doi.org/10.7554/eLife.32506.020

**Figure 3—video 3.** Dynamic distribution of MyoII visualized by sqhGFP in the wild-type hindgut epithelial cells.

DOI: https://doi.org/10.7554/eLife.32506.021

A new vertex model for a cell sheet in a 3D space, in which vertices can move in 3D, was constructed (*Figure 4A*). In this model, 452 cells (mean polygonal area of about 1) were placed in a 3D space in which the relative diameter and length of the tube reflected the in vivo situation (for the definition of the polygon area, see Materials and methods and *Figure 4—figure supplement 1*). The initial configuration of the model tube was obtained computationally (see Materials and methods). In the model, the LR asymmetry (chirality) of the hindgut epithelial cells in vivo was recapitulated qualitatively, which was achieved by anisotropic contraction of the edges (see Materials and methods) (*Taniguchi et al., 2011*). The axes of the model cell polygons tended to tilt leftward before starting the twist ($t = 0$) (*Figure 4C*), as observed in vivo (*Figure 4—figure supplement 2*). In addition, the boundaries of the model cell polygons slanted leftward ($-90$ to $0°$) more frequently than rightward ($0$ to $90°$) before starting the twist ($t = 0$) (*Figure 4D*), as previously reported in vivo (*Taniguchi et al., 2011*). In the previous study, the dissolution of cell chirality was achieved by releasing the anisotropic edge contraction (*Taniguchi et al., 2011*). Thus, we performed a relaxing procedure in the computer simulation, in which there was no anisotropic contraction of the edges ($w\alpha k = 1$, everywhere). The process of structural changes in the model hindgut is shown as still shots at $t = 1.0$ and $t = 80.0$ (*Figure 4A*). The tube had twisted $88.5°$ at $t = 80.0$, forming a left-handed screw (its tip was oriented anteriorly), which could be observed by coloring blue an array of cells that lined up straight along the anterior-posterior axis of the model gut tube before the twist and gradually became slanted during the twisting (*Figure 4A*, *Figure 4—video 1*). The tube then continued to twist more slowly ($91.0°$ at $t = 100$).

To examine the cellular dynamics associated with the hindgut tube twisting using the model, we characterized the changes in the polygon structures over time (*Figure 4C–E*). We found that most of the initial leftward tilt of the cell axes shifted to neutral as the simulation progressed (*Figure 4C*), which recapitulated the in vivo observation (*Figure 4—figure supplement 2*). This simulation also demonstrated that the dominance of leftward tilted cell boundaries at the beginning became less prominent or almost disappeared after the rotation (*Figure 4D*), which was also consistent with our

**Table 1.** Frequency of cell-intercalation events.

| Genotype | Frequency of intercalation (%) | | | n |
|---|---|---|---|---|
| | Leftward | Rightward | Total | |
| +/+ | 4.0 | 4.8 | 9.2* | 250 |
| *Myo31DF* | 2.2 | 3.3 | 6.0* | 182 |

Frequency is defined as the number of intercalation events divided by the total number of examined cells, in 30 min.

When two cells in a column had initial contact and were subsequently separated by another intervening cell, cell intercalation was said to occur, and the direction from which the intervening cell came was determined.

*includes one event that was not distinguished as leftward or rightward.

DOI: https://doi.org/10.7554/eLife.32506.022

The following source data available for Table 1:

**Table 1-Source data 1** Source data for *Table 1*.

DOI: https://doi.org/10.7554/eLife.32506.023

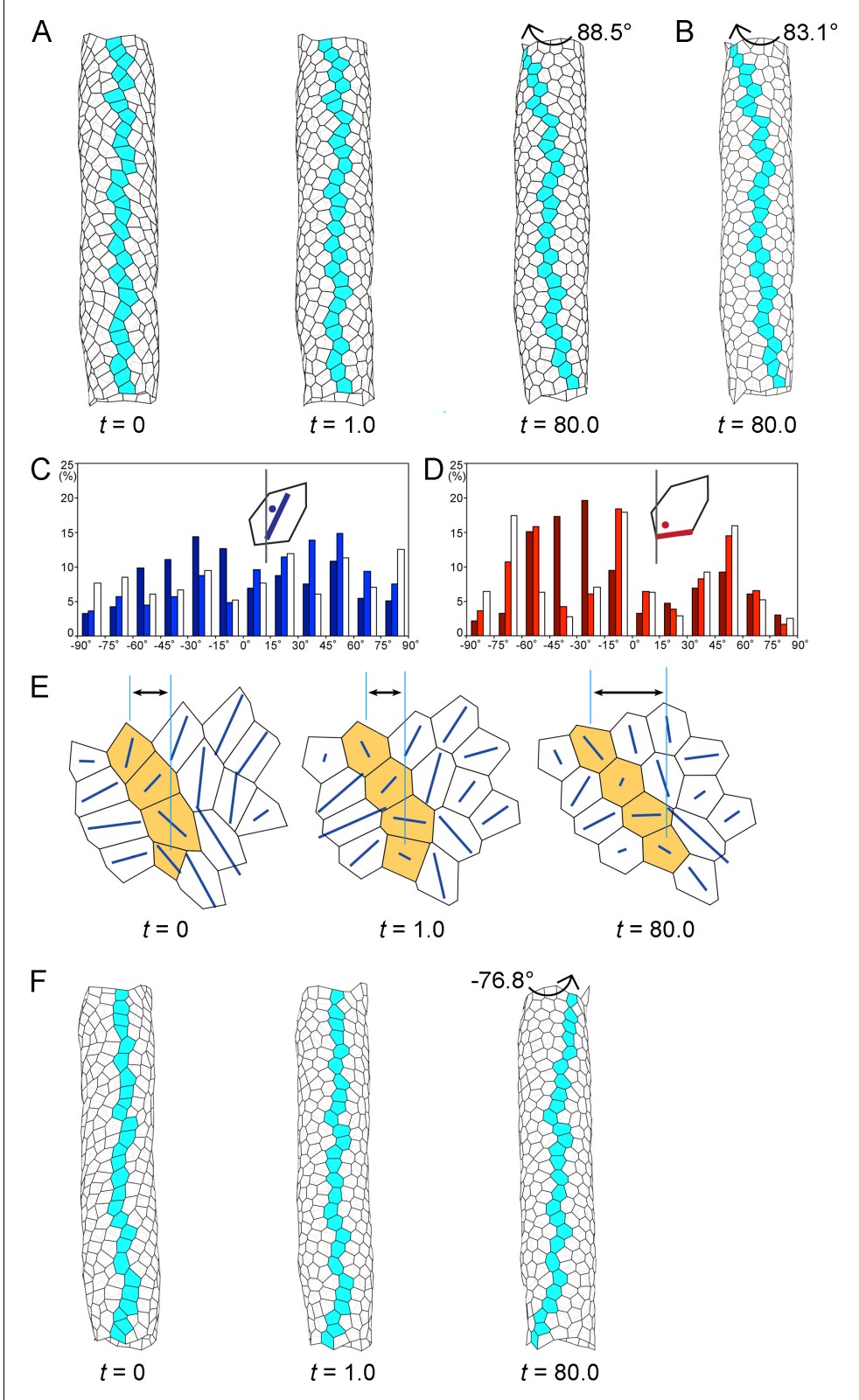

**Figure 4.** Computer simulation of the hindgut epithelial tube using a 3D cell-based vertex model for tissues and cell chirality, recapitulating the in vivo situation. (**A**) Still shots of the twisting tube in a left-handed screw (its tip is oriented anteriorly). Twist angles were 10.7° (*t* = 1.0) and 88.5° (*t* = 80.0). Cells in a line along the tube length were colored blue to show the twisting. (**B**) A computer simulation without reconnection of vertices. The model tube still rotated to a similar extent as that with reconnection. Twist angle was 83.1° (*t* = 80.0). (**C, D**) Distributions of the directions of polygon axes (**C**)

*Figure 4 continued on next page*

*Figure 4 continued*

and edges (D) in the simulation. Dark, light, and white columns denote data at *t* = 0, 1.0, and 80.0, respectively. Polygons projected on a cylindrical surface were used for this analysis as described in the Methods. Ordinate, percentage of angle distribution. Abscissa, angle θ, −90 to 90°. The axis of the polygon was the major axis of an ellipse of inertia, which was an approximation of a polygon. (E) Polygonal patterns (*t* = 0, 1.0, and 80.0) projected on a flat plane as described in Methods. An array of four polygons colored orange show a leftward tilt during *t* = 0–80.0. Blue lines show the direction of the polygon axis (the major axis of a momental ellipse). The length of the blue line shows the degree of deviation of the polygon from a circular shape. Vertical turquois lines indicate the center of the top and bottom orange-colored cells, and arrows indicate the degree of cell sliding during the rotation. (F) A model for inverted rotation in which enantiomorphic cell chirality was introduced. Twist angles were −13.5° (*t* = 1.0) and −76.8° (*t* = 80.0), respectively. In A,B,E,F '*t*' represents the number of simulation steps.

DOI: https://doi.org/10.7554/eLife.32506.024

The following video, source data, and figure supplements are available for figure 4:

**Source data 1.** Source data for *Figure 4C,D*.

DOI: https://doi.org/10.7554/eLife.32506.029

**Figure supplement 1.** Principles of our 3D cell-based vertex model for a tissue.

DOI: https://doi.org/10.7554/eLife.32506.025

**Figure supplement 2.** Cell axis distributions of the apical surfaces of hindgut epithelial cells.

DOI: https://doi.org/10.7554/eLife.32506.026

**Figure supplement 3.** Vertex model for a tube in which the vertex movement was fixed at both ends.

DOI: https://doi.org/10.7554/eLife.32506.027

**Figure supplement 4.** Vertex model for inverted rotation.

DOI: https://doi.org/10.7554/eLife.32506.028

**Figure 4—video 1.** Computer simulation of wild-type hindgut rotation using a 3D cell-based vertex model for tissues.

DOI: https://doi.org/10.7554/eLife.32506.030

**Figure 4—video 2.** Computer simulation of *Myo31DF* mutant hindgut rotation using a 3D cell-based vertex model for tissues.

DOI: https://doi.org/10.7554/eLife.32506.031

previous observation in vivo (*Taniguchi et al., 2011*). Thus, during the rotation, the cell chirality was gradually lost in the model hindgut.

Our in vivo analyses above indicated that cell sliding is a potential dynamic cell process for driving hindgut rotation. To test this possibility using the model, we investigated the detailed changes occurring in individual polygons during the twisting of the model hindgut tube, as shown in a higher magnification image of the model (*Figure 4E*). We found that in the early phases of the simulation, the polygons quickly became rounder or less polarized in shape, suggesting that the loss of chirality in the cell shape began even before the polygons started to prominently change their relative positions (*Figure 4E*, *t* = 1.0). This rapid cell deformation was also observed quantitatively, as the initial leftward biases of the cell axes (dark-colored bars, *Figure 4C*) and cell boundaries (dark-colored bars, *Figure 4D*) were largely abolished at *t* = 1.0 (light-colored bars, *Figure 4C,D*). Subsequently (*t* = 80.0), the polygons slowly changed their relative positions in a leftward direction (*Figure 4E*, *t* = 80.0). Thus, the loss of cell chirality began before the changes in the relative positions of the cells (called cell sliding in this study). During cell sliding, the cell shape change became less prominent but still continued, given that the LR biases observed at *t* = 1.0 were further randomized at *t* = 80.0 (compare light-colored and white bars, *Figure 4C,D*). These results suggested that the mechanical force was not yet balanced at *t* = 1.0, which presumably induced further events, including the cell sliding. The time lag between the initiations of cell deformation and of cell sliding suggests that these two events were mechanically distinct, probably due to the different viscoelastic properties of individual cells versus cell aggregates in our model (*Honda et al., 2004*). However, it was difficult to analyze such a time lag in vivo, because we could not simultaneously obtain high-resolution time-lapse images of nuclei and cell boundaries due to the thickness of the hindgut. Nevertheless, our 3D vertex model recapitulated chiral cell sliding that was associated with a loss of cell chirality.

Our simulation suggested that the cell chirality loss and the cell sliding are mechanically distinguishable processes. This idea was supported by another theoretical simulation, in which we fixed the vertices at the top and bottom of the tube, so the tube could not rotate (*Figure 4—figure supplement 3A–D*). In this simulation, the polygons did not slide in one direction (*Figure 4—figure supplement 3D*), while the cell chirality was still lost (*Figure 4—figure supplement 3B,C*), indicating that the cell sliding or gut-tube rotation is not required for the loss of cell chirality. These results also

showed that the loss of cell chirality and the cell sliding are not always coupled, further suggesting that they are distinct mechanical events. Therefore, this simulation further supported the idea that the mechanical force bias induced by cell chirality drives directional cell sliding and gut-tube rotation.

In the in vivo studies above, we also showed that the cell chirality and chiral cell sliding in *Myo31DF* mutants in which the direction of hindgut rotation was reversed were mirror images of their wild-type counterparts (*Taniguchi et al., 2011*). This result suggested that the initial enantio-morphism of the cell chirality determines the direction of cell sliding, which consequently defines the direction of the hindgut rotation. To verify this idea, we performed another simulation recapitulating the inverted rotation of the model gut tube. In this simulation, we introduced the enantiomorphic chirality into the 3D vertex model, and otherwise the same parameters were used as described above. We found that the inverted rotation and cell sliding were recapitulated in this simulation (*Figure 4F*, *Figure 4—figure supplement 4*, *Figure 4—video 2*). These results supported the idea that the initial cell chirality determines the direction of cell sliding, as predicted in the in vivo studies above.

Our in vivo studies suggested that cell intercalation does not play a major role in the LR directional rotation of the hindgut. Consistent with this idea, we found that cell intercalation rarely occurred during the simulation, although our model allowed for it [it occurred only 12 times for 452 vertices (2.7%) by $t = 80$; for example, we did not observe any cell intercalation among the array of blue colored cells (*Figure 4A*). The frequency of cell intercalation in vivo (9.2% every 30 min for 60 min) was much higher than that of our vertex model. This discrepancy was probably because our model did not include the anterior-posterior elongation (convergent extension) of the hindgut, which occurs in vivo at the same time as the hindgut rotation (*Lengyel and Iwaki, 2002*). Cell intercalation is known to be required for the convergent extension of the hindgut (*Johansen et al., 2003*). To confirm that cell deformation rather than cell intercalation is a major driving force of the cell sliding and the hindgut rotation, we generated a computer simulation that did not allow junctional remodeling (*Figure 4B*). In this simulation, the cells still slid and the tube rotated (twist angle 83.1° at $t = 80$). In this respect, the cell sliding might be considered a type of autonomous shear deformation at the multicellular level, although it is not directly induced by the loss of chirality in the shape of each individual cell or by a simple LR asymmetric elongation of the cell. Thus, collectively, the simulations indicated that cell sliding induced by a cell-chirality-provoked mechanical force bias is a dynamic cellular process connecting cell chirality and LR asymmetric tube twisting. In conclusion, our in vivo and in silico analyses both demonstrated that cell sliding converts the intrinsic chirality of the cell shape into the LR asymmetric epithelial morphogenesis though chiral cell deformation.

## Discussion

### Chiral cell sliding in LR asymmetric rotations of organs

The morphogenesis of epithelial cell sheets in the absence of cell number change has been explained as a consequence of cell-shape changes and/or cell rearrangements. Regarding cell rearrangements, the cell intercalation in convergent extension, in which cells intercalate in a medial direction to induce cell sheets to elongate in an anterior-posterior direction, is well described (*Bertet et al., 2004*; *Honda et al., 2008*; *Uriu et al., 2014*). Directional cell intercalation is also important in the LR asymmetric rotation of the male genitalia in *Drosophila* (*Sato et al., 2015a*). However, in the present study, we revealed that the LR asymmetric rotation of the embryonic *Drosophila* hindgut is achieved by chiral cell sliding, in which cells change their relative position against their subjacent neighbors in one direction. In contrast to directional cell intercalation, this cell sliding does not necessarily require cell rearrangements (*Figure 3A,B*). Indeed, we showed that a computer simulation in which cell-boundary remodeling was prohibited still rotated as well as our standard 3D vertex model hindgut (*Figure 4B*). Thus, in the hindgut case, we speculate that the accumulation of small chiral cell sliding events between adjacent cells in the hindgut epithelial tube are sufficient to induce the 90° rotation of the hindgut epithelial tube, because of the large difference between the radius and the height of this organ.

Considering that cell sliding and cell rearrangement may not be mutually exclusive, the definition of cell sliding may be extended. For example, theoretical models of *Drosophila* male genitalia

rotation include the sliding motion of cells (*Sato et al., 2015a*, *2015b*), although this mechanism also requires cell rearrangements, because a large amount of cell displacement takes place. In any event, for the dynamic cell events induced by cell chirality, cell sliding appears to drive the LR asymmetric rotation of tissues with or without cell intercalations. Thus, cell sliding may be a commonly used mechanism in LR asymmetric morphogenesis, although it has not been demonstrated to date.

Directional cell intercalation often requires a biased subcellular distribution of Myosin II (*Bertet et al., 2004*; *Sato et al., 2015a*). For example, during the LR asymmetric rotation of *Drosophila* genitalia, LR asymmetric cell intercalation is induced by the LR asymmetric distribution of Myosin II (*Sato et al., 2015a*). However, in the *Drosophila* hindgut, a biased Myosin II distribution was not detected in fixed (*Hatori et al., 2014*; *Taniguchi et al., 2011*) or live tissues (this study). Given that chiral cell sliding requires a very small amount of change in the relative positions of cells, a large bias in the amount of Myosin II may not be required. Indeed, our simulation revealed that the anisotropic contraction of edges before but not during the rotation was sufficient to induce whole-organ rotation (*Figure 4A*), whereas the anisotropic contraction of edges needs to be maintained in the computer simulation to recapitulate the rotation of *Drosophila* male genitalia (*Sato et al., 2015a*). Thus, if there is any bias in the subcellular distribution of Myosin II, it might be too subtle to detect by the methods we used (*Hatori et al., 2014*; *Taniguchi et al., 2011*). Alternatively, force-generating mechanisms other than the biased expression of Myosin II might be used in the LR asymmetric rotation of the hindgut.

## Cell chirality causes LR asymmetric organogenesis

In our computer simulation, we recapitulated the LR directional axial rotation of the gut tube, which is driven by a mechanical force bias induced by cell chirality. Our analyses of the model cells in the simulations suggested that the rotation of the model gut tube is achieved through two steps: the first is a deformation in the shape of each cell (loss of cell chirality) without a change in the relative positions of anterior-posterior neighboring cells, and the second is a change in the relative positions of anterior-posterior neighboring cells (called cell sliding in this study), which presumably drives the rotation of the model gut tube. Although the cell deformation and cell sliding temporally overlapped in part (*Figure 4A,C–E*), the following observations suggested that these two events are mechanically distinct. First, the initiation of cell chirality loss mostly preceded the onset of cell sliding (*Figure 4A,C–E*). The timing of the model gut tube rotation coincided with that of the cell sliding (*Figure 4A,E*), but not of the cell chirality loss (*Figure 4C–E*), suggesting that the rotation of the model gut tube involves two cellular dynamic steps. Second, in a simulation prohibiting the tube rotation, cell sliding did not occur, although the loss of cell chirality was still observed (*Figure 4—figure supplement 3*), indicating that the loss of cell chirality is not always coupled with cell sliding. Based on these results, we speculated that the loss of cell chirality and the cell sliding are distinct mechanical processes.

The differences between the cell chirality loss and cell sliding could be explained by their distinct mechanical properties. In our simulation, the equalization of the initial anisotropic contractile force was the only source driving subsequent events. During the balancing processes, individual cells were deformed quickly; most of the deformation was achieved by $t = 1.0$, although the force balance had not yet been neutralized given that the deformation continued to $t = 80.0$ (*Figure 4C,D*). In vertex models, a viscoelastic property is generally reflected in the speed of the response upon a force application (*Honda et al., 2008*, *2004*). Thus, we speculate that changes in the relative positions of cells (cell sliding) slowly occurred after the major dissolution of cell chirality, due to the difference in viscoelastic properties of the individual cell versus the cell aggregate, until the initial anisotropy of the force was neutralized. In turn, completion of the cell sliding led to global stabilization of the model tube.

Our conclusions about the mechanical properties of chiral cell sliding were based on the computer simulation. However, our ideas did not contradict the observations made in the embryonic hindgut in vivo. In this study, we also found that the hindgut itself generates the active force for driving its rotation. Considering that cell chirality was observed before the hindgut rotation and was lost during the rotation in vivo, the cancelation of cell chirality is probably a primary driving force for the subsequent events including the cell sliding and hindgut rotation, in vivo. This result also excludes the possibility that the cell sliding is a consequence of hindgut rotation driven by an external force, further supporting the validity of our computer simulation for analyzing the relationship between cell

chirality loss and cell sliding. In addition, the enantiomorphism of the initial cell chirality is correlated with the LR direction of the cell sliding, based on our analyses of the LR inversion mutant, *Myo31DF* (*Figure 2D*). Thus, the LR direction of the cell sliding appears to depend on the initial cell chirality. These results are consistent with the conclusions obtained from our computer simulations. In conclusion, we propose that chiral cell sliding is a cell dynamic mechanism that connects cell chirality with the LR asymmetric rotation of the hindgut tube.

## Prospective roles of cell chirality based on its evolutionary conservation

In vertebrates, organ laterality is thought to be determined by the LR body axis established by Nodal cassette gene expressions, which is distinct from the mechanisms reported for *Drosophila* (*Huang et al., 2014*; *Yoshiba and Hamada, 2014*). However, a recent study revealed that LR asymmetry formation in the zebrafish heart does not depend on *nodal* gene expression (*Noël et al., 2013*). Moreover, explanted linear hearts of various species develop dextral looping in culture (*Bacon, 1945*; *Manning and McLachlan, 1990*; *Noël et al., 2013*), suggesting that the heart LR asymmetry formation is tissue-intrinsic. In *Drosophila*, which lacks the Nodal cassette gene functions, various organs use cell chirality as a common strategy to establish LR asymmetry in a tissue-intrinsic manner, in the absence of an LR body axis (*González-Morales et al., 2015*; *Sato et al., 2015a*; *Taniguchi et al., 2011*; *Vorbrüggen et al., 1997*). To date, cell chirality in tissues has been observed only in *Drosophila* (*González-Morales et al., 2015*; *Sato et al., 2015a*; *Taniguchi et al., 2011*). However, cell chirality has been reported in cultured cells, including those of vertebrates (*Chen et al., 2012*; *Raymond et al., 2016*; *Wan et al., 2011*; *Xu et al., 2007*). Importantly, many cell types from various organs show a chiral cell shape (*Chen et al., 2012*; *Raymond et al., 2016*; *Wan et al., 2011*). Therefore, cell chirality may play specific roles in the LR asymmetric organogenesis in vertebrates as well.

# Materials and methods

**Key resources table**

| Reagent type (species) or resource | Designation | Source or reference | Identifiers | Additional information |
|---|---|---|---|---|
| Strain, strain background (*D. melanogaster*) | *byn*-Gal4 | *Iwaki and Lengyel (2002)*, PMID:12175491 | | |
| Strain, strain background (*D. melanogaster*) | UAS-*redstinger* | Bloomington Drosophila Stock Center | BDSC Cat# 8547, RRID:BDSC_8547 | |
| Strain, strain background (*D. melanogaster*) | UAS-*stinger* | Bloomington Drosophila Stock Center | BDSC Cat# 28845, RRID:BDSC_28845 | |
| Strain, strain background (*D. melanogaster*) | UAS-*myrGFP* (JFRC29) | *Pfeiffer et al. (2012)*, PMID: 22493255 | | |
| Strain, strain background (*D. melanogaster*) | UAS-*NLS-tdTomato* | this study | | An NLS-tdTomato fragment obtained from pQC NLS TdTomato IX (Addgene, #37347) was cloned into 20 × UAS vector. Insertion site is attP2. |
| Strain, strain background (*D. melanogaster*) | UAS-*sqhGFP* | this study | | A sqhGFP fragment obtained from a sqhGFP fly strain (BL 57144) was cloned into pUAST (*Brand and Perrimon, 1993*). Random insertion. |
| Strain, strain background (*D. melanogaster*) | *Myo31DF*$^{L152}$ | *Hozumi et al. (2006)*, PMID: 16598258 | | |

## Live imaging

Dechorionated *Drosophila* embryos were placed on grape juice agar plates. Late stage 12 embryos of the appropriate genotype were selected under florescence microscopy and mounted dorsal side up on double sticky tape on slide glasses. We added oxygen-permeable Halocarbon oil 27 (Sigma), and overlaid a coverslip of regular thickness over the embryos using 0.17–0.25 mm-thick coverslips as spacers. We imaged embryos every 5 min for 2 hr with a scanning laser confocal microscope, LSM

700 (Zeiss) or A1 (Nikon) at 22–25°C. The fly strains used were *byn*-Gal4 (*Iwaki and Lengyel, 2002*), UAS-*redstinger* (BL8547) (*Barolo et al., 2004*), UAS-*stinger* (BL28845), UAS-*myrGFP* (JFRC29) (*Pfeiffer et al., 2012*), UAS-*NLS-tdTomato*, UAS-*sqhGFP* (see below), and *Myo31DF^{L152}* (*Hozumi et al., 2006*).

## Quantitative analysis of cell movement and cell deformation

For the cell movement analysis, we tracked the position of the cell nucleus manually using ImageJ software and measured the x,y coordinates every 30 min. We also used a particle analysis plugin for tracking, and obtained comparable results. We only analyzed the three central columns of cells in the root part of the hindgut, to minimize the influence of the tubular hook shape of the hindgut on the displacement measurements. In the analysis, we set the subjacent cell position as (0,0) and measured the relative displacement of the upper cell in the x direction (*Figure 1E*). Statistical analyses were performed using Student's *t*-test. For the cell deformation analysis, we measured the angle changes in the boundaries between cells aligned in a column every 30 min. Statistical analyses were performed using the $\chi^2$ test. For the cell intercalation analysis, we used both the cell nucleus and boundary images. We defined cell intercalation as two cells that had initial contact and were separated by another cell within 30 min. We calculated the frequency by dividing the number of intercalation events by the total number of examined cells. We also examined the direction from which the intervening cell came and the angles of the diminishing cell boundaries. Statistical analyses were performed using the $\chi^2$ test.

## Computer simulation

Summary: The model tube consisted of 452 polygonal cells. The polygons covered the surface of the tube without overlaps or gaps. The height of the tube was 28.7, its diameter was 5.0, and the mean polygonal area was about 1. The cell number and ratio of the height to diameter reflected those of the hindgut in vivo. The polygonal pattern of the surface was described by the positions of vertices (*x-*, *y-*, *z*-coordinates) of the polygons. Movements of the vertices were calculated by differential equations, such as the cell-based vertex dynamics for tissues, which has a potential energy term (see below). The potential energy term mainly involved the edge energy and elastic energy of the polygonal area. According to vertex dynamics, when the vertices moved, the edge energy and the surface elastic energy were reduced. We added another process to the system of differential equations: when the edge length became less than a critical length δ, a reconnection of vertices took place causing a change in the polygon shape (*Figure 4—figure supplement 1*). Thus, we could obtain a stable state of the tube using computer simulations. Numerical calculations of the differential equations were performed using the Runge-Kutta method with step size *h* (=0.005).

Initial model tube for computer simulations including cells with chiral properties: We made an initial tube recapitulating the in vivo hindgut, in which the cell axes tilted leftward as shown in *Figure 4C*, as follows. In a rectangular area (15.75 × 28.7) we distributed 452 circular dishes (diameter = 0.82) at random with the periodic boundary condition, and performed Dirichlet (or Voronoi) tessellation (*Honda, 1978*). The rectangle was converted into a cylinder in 3D space as described by *Honda et al. (2008)*, resulting in a model tube consisting of 452 polygons.

For each cell to have chiral properties, we introduced anisotropic contraction of the edges of the polygonal cells. Each cell had its own polarity, which was defined as a deflection angle from the anterior-posterior axis (AP axis) of the system at the initial stage of the simulations. That is, we considered a plane (blue rectangle) that was perpendicular to the polarity direction (blue arrow) and included the central point of the polygon (*Figure 4—figure supplement 1A*). The plane crossed two edges of the polygon, and the two edges (expressed by thick black lines) were the specific edges undergoing the strongest contraction. Thus, the polarity determined two special edges that were approximately parallel to the polarity direction. The special edges had a high energy density ($w_{\alpha k}$, $w_{\alpha k}$-fold higher than that of the other edges) and contracted more strongly. The polygons then became elongated and assumed a chiral shape. To give chiral properties to the polygons for which the polygon axes tilted leftward, we applied vertex dynamics to the model tube, where the deflection angle was +30° and weight was applied to the strongly contracting edges, $w_{\alpha k}$3.5. The vertex dynamics stopped at *t* = 5.0. As a result, we obtained a tube as described in *Figure 4A* (*t* = 0).

For the computer simulation of inverted twisting (*Figure 4F*, *Figure 4—figure supplement 4*), we used another initial tube as shown in *Figure 1E* (t = 0), by similar methods except that the deflection angle was −35° and t = 10.0. The parameter values were not exactly enantiomorphic between the two initial tubes, because the original tube from which the two initial tubes were made was not perfectly symmetric. For the computer simulation of stopped rotation (*Figure 4—figure supplement 3*), we used the same initial tube as we used for the wild-type chirality (see *Figure 4A*, t = 0). In this case, the rotation of the vertices at the top and bottom of the tube was fixed; that is, the x- and y-coordinates of the vertices were fixed.

Vertex dynamics for a sheet in 3D space: A tube composed of multiple cells was considered as a curved sheet consisting of polygons without gaps or overlaps. The edges (boundaries between cells) and area of the polygons (cell volume) were expressed as x-, y-, and z-coordinates of the vertices (*Figure 4—figure supplement 1B*). The spatial relationships between neighboring vertices were defined by the surrounding polygons (*Honda et al., 2004*; *Nagai and Honda, 2001*). The vertices obeyed the equation of motion:

$$\eta \, \mathrm{d}r_i/\mathrm{d}t = -\nabla_i U \qquad (i = 1, ..., n_v) \tag{1}$$

where $r_i$ is a 3D-positional vector of vertex $i$, $\nabla_i$ is the nabla differential operator, and $n_v$ is the total vertex number ($n_v$ = 904). The left side of *Equation (1)* represents a viscous drag force proportional to the vertex velocity $\mathrm{d}r_i/\mathrm{d}t$ with a positive constant $\eta$ (an analog of the coefficient of viscosity). Vertices do not have mass (inertia), so the motion of the vertices and polygons is completely damped. Differentiation of the potential $U$ with respect to time $t$ yields the following inequality using *Equation (1)*,

$$\mathrm{d}U/\mathrm{d}t = \Sigma_i \nabla_i U \mathrm{d}r_i/\mathrm{d}t = -\eta \, \Sigma_i \, (\mathrm{d}r_i/\mathrm{d}t)^2 \le 0. \tag{2}$$

The inequality indicates that vertices move to decrease $U$ (strictly, not to increase $U$). We obtained a stable shape described by vertices using *Equation (1)*.

The right side of *Equation (1)* represents a potential force (driving force), that is, minus the gradient of the potential $U$. The potential $U$ includes various terms related to the edges and surface areas of polygons and the tube volume, which are all expressed by vertex positions. Therefore, $U$ is a function of the vertex coordinates.

In the present study, the potential $U$ contains terms for edge energy ($U_L$), elastic surface energy ($U_{ES}$), elastic volume energy ($U_{EV}$), and the boundary restriction energy of the top and bottom of the tube ($U_B$):

$$U = U_L + U_{ES} + U_{EV} + U_B. \tag{3}$$

The potential $U_L$ denotes the total edge energy of the cells:

$$U_L = \sigma_L \Sigma^n_{\ a} \left( \Sigma^{n_a}_{\ k} w_{\alpha k} L_{\alpha k} \right). \tag{4}$$

The term ($\Sigma^{n_\alpha}_{\ k} w_{\alpha k} L_{\alpha k}$) in *Equation (4)* represents the edge energy of cell α. $n_\alpha$ is the edge number of cell α. $L_{\alpha k}$ and $w_{\alpha k}$ are the edge length and weight applied to the respective edge. The weight, $w_{\alpha k}$, depends on the edge species. When an edge contracts strongly, $w_{\alpha k}$ is large (=1.3 or 2.0); otherwise, $w_{\alpha k}$k1.0. αk designates the $k$-th edge of polygon α. $n$ is the total polygon number except for the top and bottom polygons ($n$ = 452). $\sigma_L$ is the edge energy density.

The potential $U_{ES}$ denotes the total elastic energy of the polygon area:

$$U_{ES} = \kappa_S \Sigma^n_{\ a} \left( S_a - S_o \right)^2, \tag{5}$$

where $S_\alpha$ and $S_o$ are the polygon area at time $t$ and the polygon area at the relaxed state, respectively. $\kappa_S$ is the elastic energy density of the polygon area.

The potential $U_{EV}$ denotes the elastic energy of the tube:

$$U_{EV} = \kappa_V \left( V_a - V_o \right)^2, \tag{6}$$

where $V_\alpha$ and $V_o$ are the tube volume at time $t$ and the tube volume at the relaxed state, respectively. $\kappa_V$ is the elastic energy density of the tube volume.

The potential $U_B$ denotes the boundary restriction energy of the top and bottom of the tube:

$$U_B = \kappa_B \Big\{ \Sigma_j^{nvTop} [(r_j - r_{Top})^2 - R_{Top}^2]^2$$
$$+ \Sigma_j^{nvBottom} [(r_j - r_{Bottom})^2 - R_{Bottom}^2]^2 \Big\}. \tag{7}$$

Centers of the top and bottom polygons of the tube are $r_{Top}$ and $r_{Bottom}$, respectively. The vertices of the top and bottom polygons of the tube are restricted to the circles of the top and bottom ends of the tube (radii are $R_{Top}$ and $R_{Bottom}$, respectively). $\kappa_B$ is the elastic constant of the circular array of vertices of the top and bottom polygons.

Thus, *Equation (1)* takes the form:

$$\eta dr_i/dt = -\nabla_i \Big\{ \sigma_L \Sigma_a^n (\Sigma_k^{na} w_{ak} L_{ak})$$
$$+ \kappa_S \Sigma_a^n (S_a - S_o)^2 + \kappa_V (V_a - V_o)^2$$
$$+ \kappa_B \Sigma_j^{nvTop} \Big[ (r_j - r_{Top})^2 - R_{Top}^2 \Big]^2$$
$$+ \kappa_B \Sigma_j^{nvBottom} \Big[ (r_j - r_{Bottom})^2 - R_{Bottom}^2 \Big]^2 \Big\}. \tag{8}$$

To reduce the parameter number without a loss of generality, we introduced a new length unit $R_o$ and rewrote *Equation (8)* using new dimensionless quantities $r_i$ as follows:

$$r_i = r_i{''} R_o, \nabla_i = \nabla_i{''}/R_o, L_{ak} = L_{ak}{''} R_o,$$
$$S_a = S_a{''} R_o^2, S_o = S_o{''} R_o^2, V = V{''} R_o^3,$$
$$V_o{''} R_o^3, r_{Top} = r_{Top}{''} R_o, r_{Bottom} = r_{Bottom}{''}$$
$$R_o, R_{Top} = R_{Top}{''} R_o, R_{Bottom} = R_{Bottom}{''} R_o. \tag{9}$$

Thus, *Equation (8)* takes the form:

$$dr_i{''}/dt{''} = -\nabla_i{''} \Big\{ \sigma_L{''} \Sigma_a^n (\Sigma_k^{na} w_{ak}{''} L_{ak}{''}) + \kappa_{S{''}} \Sigma_a^n (S_a{''} - S_o{''})^2$$
$$+ \kappa_V{''} (V{''} - V_0{''})^2 + \kappa_{B{''}} \Sigma_j^{nvTop} [(r_j{''} - r_{Top}{''})^2 - R_{Top}{''2}]^2$$
$$+ \kappa_{B{''}} \Sigma_j^{nvBottom} [(r_j{''} - r_{Bottom}{''})^2 - R_{Bottom}{''2}]^2 \Big\}. \tag{10}$$

in which the new quantities are defined as follows:

$$t{''} = t/(\eta R_o), \sigma_L{''} = \sigma_L, w_{ak}{''} = w_{ak}, \kappa_S{''} = \kappa_S R_o^3,$$
$$\kappa_V{''} = \kappa_V R_o^5, \kappa_B{''} = \kappa_B R_o^3. array \tag{11}$$

We take $R_o$ ($=S_o^{1/2}$)=1, so that $S_o = 1$. *Equation (10)* lacks explicit parameters corresponding to $\eta$. Thus, without a loss of generality, we can describe cell behaviors using the simple parameters $\kappa_S$, $\kappa_V$, and $\kappa_B$. Below, the cell motions were measured in terms of the new length unit $R_o = S_o^{1/2}$ and the new time unit $1/(\eta R_o)$, which are the characteristic length scale and time scale of the tube, respectively.

Hereafter, we omit the primes ('') on the rescaled quantities in *Equation (9)*.

$$dr_i/dt = -\nabla_i \Big\{ \sigma_L \Sigma_a^n (\Sigma_k^{na} w_{ak} L_{ak}) + \kappa_S \Sigma_a^n (S_a - S_o)^2$$
$$+ \kappa_V (V - V_0)^2 + \kappa_B \Sigma_j^{nvTop} [(r_j - r_{Top})^2 - R_{Top}^2]^2$$
$$+ \kappa_B \Sigma_j^{nvBottom} [(r_j - r_{Bottom})^2 - R_{Bottom}^2]^2 \Big\}. \tag{12}$$

We used the parameter values $\sigma_L = 2.2$, $\kappa_S = 10.0$, $\kappa_V = 0.2$, $\kappa_B = 1.0$, $R_{Top} = R_{Top} = 2.5$, and $V_o = 566.64$ to obtain a stable shape of the tube.

Elementary process of reconnecting neighboring vertices: In addition to the equations of motion, our model involves an elementary process of reconnecting neighboring vertices (*Honda et al., 2004*; *Nagai and Honda, 2001*). We extended the reconnection into the 3D space as shown in *Figure 4—figure supplement 1C*. When the length of an edge connecting two neighboring vertices

became shorter than a critical length δ(=0.3), the relationship of the neighboring vertices changed and the neighbors reconnected with each other.

We also performed a computer simulation that did not allow the reconnection of vertices to test whether cell intercalation is required for the hindgut rotation (*Figure 4B*).

### Shape analysis of polygons

Polygons on the tube surface were projected onto a geometrical cylinder (diameter of 5.0), and the surface of the cylinder was extended to a flat plane. To analyze the directions of cell axes and edges statistically, projected polygons, except for polygons close to the peripheral parts of the tube, were used. To determine the polygon shape, a polygon was approximated by a momental ellipse (ellipse of inertia), and the direction of the major axis of the ellipse was defined as the direction of the polygon. The degree of deviation from the circle was defined as $(d_{max} - d_{min})/(d_{max} + d_{min})$, where $d_{max}$ and $d_{min}$ were the lengths of the major and minor axes of the ellipse, respectively.

### Explant culture of hindgut

The culture medium consisted of equal volumes of M3 medium and fly extract. Embryos expressing UAS-*GFP-moesin* (*Chihara et al., 2003*) driven by *en*-Gal4 (Bloomington Stock Center) were dechorionated using double-sided tape. In the hindgut, UAS-*GFP-moesin* is specifically expressed in the dorsal side, which allowed us to determine the direction of hindgut rotation. The hindgut was dissected with a tungsten needle and mounted in the medium. Images were captured using an LSM 5 PASCAL laser-scanning microscope (Zeiss).

### Transgenic flies

We generated transgenic lines carrying UAS-*NLS-tdTomato* and UAS-*sqhGFP*. For UAS-*NLS-tdTomato*, we first made a vector with 20 × UAS by inserting a 10 × UAS fragment amplified from pJFRC81 (*Pfeiffer et al., 2012*) by PCR using the primers AAAGCTAGCTCAACGACAGGAGCACGATC and AAAGACGTCTCAACGACAGGAGCACGATC, into pJFRC81. An NLS-tdTomato fragment (NotI-NheI) from pQC NLS TdTomato IX (Addgene, #37347) was inserted into the NotI and XbaI sites of the vector. For UAS-*sqhGFP*, a *sqhGFP* fragment was amplified by PCR from genomic DNA isolated from a *sqhGFP* fly strain (BL 57144) using the primers CACCGCGGCCGCATGTCATCCCGTAAGACCGC and CACCGGTACCCTATTTGTATAGTTCATCCA, and cloned into the NotI and KpnI sites of pUAST (*Brand and Perrimon, 1993*). To generate transgenic lines, we used the attP2 site for UAS-NLS-tdTomato and random insertion for UAS-sqhGFP. For imaging analyses, flies carrying UAS-sqhGFP insertions in the second and third chromosomes were used.

### Videos

To track the movement of nuclei, we adjusted the depth of the z stack or z slice at each time point to make clear time-lapse movies, because the hindgut cells change their z position over time.

## Acknowledgements

We thank Gerald Rubin and the Bloomington Stock Center for providing the fly strains. We also thank the Advanced Center for Computing and Communication (ACCC), RIKEN, Wako, Japan and the Institute of Statistical Mathematics (ISM), Tokyo, Japan for their supercomputing facilities. This work was supported by JSPS KAKENHI Grants #JP25440117 to HH., #JP17K07410 to HH., #JP15H05863 to KM., and #JP15K07077 to M Inaki.

## Additional information

### Funding

| Funder | Grant reference number | Author |
| --- | --- | --- |
| Japan Society for the Promotion of Science | JP15K07077 | Mikiko Inaki |

| Japan Society for the Promotion of Science | JP15H05863 | Kenji Matsuno |
| Japan Society for the Promotion of Science | JP25440117 | Hisao Honda |
| Japan Society for the Promotion of Science | JP17K07410 | Hisao Honda |

The funders had no role in study design, data collection and interpretation, or the decision to submit the work for publication.

### Author contributions

Mikiko Inaki, Conceptualization, Funding acquisition, Validation, Methodology, Writing—original draft, Project administration, Writing—review and editing, Acquisition of data, Data analysis; Ryo Hatori, Conceptualization, Methodology, Acquisition of data, Data analysis; Naotaka Nakazawa, Methodology, Acquisition of data, Data analysis; Takashi Okumura, Tomoki Ishibashi, Methodology, Acquisition of data; Junichi Kikuta, Resources, Methodology, Acquisition of data; Masaru Ishii, Conceptualization, Resources, Project administration, Writing—review and editing; Kenji Matsuno, Conceptualization, Funding acquisition, Writing—original draft, Project administration, Writing—review and editing; Hisao Honda, Conceptualization, Validation, Methodology, Writing—original draft, Project administration, Writing—review and editing, Acquisition of data, Data analysis

### Author ORCIDs

Kenji Matsuno http://orcid.org/0000-0001-6140-8833

### Decision letter and Author response

Decision letter https://doi.org/10.7554/eLife.32506.034
Author response https://doi.org/10.7554/eLife.32506.035

## Additional files

### Supplementary files

• Transparent reporting form
DOI: https://doi.org/10.7554/eLife.32506.032

### Data availability

All data generated or analysed during this study are included in the manuscript and supporting files. Source data files have been provided for Figures 1, 2, 3 and 4.

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
