## [Decision Letter]

Thank you for submitting your article "Chiral cell sliding drives left-right asymmetric organ twisting" for consideration by *eLife*. Your article has been reviewed by three peer reviewers, and the evaluation has been overseen by a Reviewing Editor and K VijayRaghavan as the Senior Editor. The reviewers have opted to remain anonymous.

The reviewers have discussed the reviews with one another and the Reviewing Editor has drafted this decision to help you prepare a revised submission.

Summary:

The authors have studied the cellular dynamic mechanism underlying left-right asymmetric organ twisting, in particular, how cellular chirality is converted into axial rotation of the *Drosophila* hindgut. They first constructed a 3D vertex model that can recapitulate in vivo cellular behaviors. Analysis of the model suggested a novel mechanism in which cells directionally change their positions relative to their neighbors without cell intercalation, which they termed "cell sliding". As predicted by the model, cell sliding was observed in hindgut epithelial cells that undergo axial rotation, and the direction of cell sliding was reversed in a mutant *Drosophila* with hindgut rotation inversed. In all, the current results suggest that cellular chirality, which already exists in individual cells of the hindgut, will induce directional cell sliding, which in turn result in the hindgut twisting. This is a beautiful study combining mathematical modeling and experimental biology, and cell sliding is a new concept for cell biology in general (but see cautionary comments below!)

Essential revisions:

1) The manuscript by Inaki and colleagues propose a new cellular mechanism "cell sliding" that drives tissue morphogenesis in *Drosophila* embryonic hindgut rotation. This is one of the follow-up papers from the same lab, which showed that this hindgut rotation is driven by the cell chirality, more specifically the anisotropic contractility of cell boundaries (Taniguchi et al., 2011).

In this manuscript, the authors first use a 3D cell-base vertex model, and show that the cells lost their chirality before tissue rotation (or cell sliding). The authors' reasons that this temporal difference between the cancellation of cell chirality and the cell sliding arise from the viscoelastic nature of the tissue. The authors then present the experimental data characterizing the "cell sliding". By tracking the position of the nucleus and the angle of the cell boundary, the authors show that the cells translocate their position during the hindgut rotation. Furthermore, the authors find that the cell boundary associates with cell interaction is rare and that most of the boundaries rotate its angle without junctional remodeling. In addition to the analyses of wild-type case, the authors perform a computational simulation, live imaging, and analyses for Myo31F mutant, which exhibit reverse hindgut rotation.

The evidence to claim that the "cell sliding" is an active process, which drives tissue rotation, is weak. The major data the authors put forward to claim this point is the in silico results showing the cancellation of cell chirality precedes the cell sliding (and tissue rotation). Based on this observation, the authors argue that the cell sliding is caused by the cancellation of cell chirality. This claim raises a few important questions. What would be the mechanisms (or driving force) of cell sliding (and tissue rotation) under the circumstance where the cell chirality is neutralized before the rotation? If the tissue rotation is a result of the viscoelastic response of the tissue after "relaxing procedure", then how this response cause only the cell sliding during tissue rotation but not the cell chirality change? Moreover, it is not clear whether this time lag between the cancellation of cell chirality and the cell sliding can be observed in *Drosophila* embryonic hindgut.

Overall, the current manuscript is not sufficient to reject the possibility that the cell sliding observed here was a consequence of tissue rotation, i.e., a passive process. The authors need to carefully discuss these contrary possibilities.

2) The term "cell sliding" can be misleading, since, for most cases, the adjacent cells still share the same boundary without moving away from each other. Cell sliding needs a clear definition. Is it the cell shape change? Even if we can exclude the possible role of cell intercalation, it is not clear how the authors reach the conclusion that it is chiral cell sliding that drives asymmetry. What drives the chiral cell sliding then? I suppose that it is planar cell shape chirality. How do the two differ from each other?

3) The section on chiral properties of cells doesn't seem sufficient to induce a chiral bias. The determination of special edges needs to be biased towards a specific direction to induce asymmetry. Also, how is the deflection angle defined? Was it used as an input for the simulation?

4) In the numerical simulation, the neighbor exchange is enabled. What if it is disabled? Will the asymmetric looping be still possible? What will happen to the angle of twisting?

5) Could cell intercalation be possibly a mechanism for amplifying the asymmetry? Will all cells tilted, and the intercalation can increase the total length without inducing too much stretching/elongation of the cells?

6) There are significant concerns regarding the organization of this manuscript. The Table 1 and Figure 3—video 3 may be used as a motive for the entire study, instead of merely mentioning them towards the end. It may be more reasonable to follow the table and video with experimental findings on cell sliding and end with numerical simulation.

[Editors' note: further revisions were requested prior to acceptance, as described below.]

Thank you for resubmitting your work entitled "Chiral cell sliding drives left-right asymmetric organ twisting" for further consideration at *eLife*. Your revised article has been favorably evaluated by K VijayRaghavan (Senior Editor), a Reviewing Editor, and two reviewers.

The manuscript has been improved but there are some remaining issues that need to be addressed before acceptance, as outlined below. Along with responding to these comments from the reviewers, please expand your explanations/interpretations of the data and the speculations of the potential mechanisms. This doesn't require any additional experiments or simulations.

*Reviewer #1:*

This manuscript studied cellular biophysics involving in the left-right asymmetric twisting of *Drosophila* embryonic hindgut. They developed a 3D tubular model for studying gut asymmetric rotation, and proposed that cell asymmetric sliding instead of cell intercalation determines the directionality of gut looping. Overall, the research topic is important and interesting. With the revision, the quality of manuscript has largely improved. However, there are still some concerns for the authors to consider:

1) One confusing part of the manuscript is regarding the relationships between cell deformation, cell sliding, cell chirality (and its loss) and tissue rotation. This is also reflected in previous comments raised by the reviewers and editor. It is partially due to the adopted definition of cell chirality in the manuscript, which is defined as the directional biases of major axes, or cell alignment (Figure 3). However, based on this definition, cell alignment direction geometrically changes with individual cell deformation, which drives cell sliding at a multicellular level. Therefore, all these three are coupled with one another, and, in my opinion, that based on the current definition of "observed cell chirality", it is impossible to distinguish between them or tell which ones occur first and later.

*Reviewer #3:*

Overall, the author responded to the reviewers’ comments nicely in the rebuttal. However, I personally feel that many of the comments were not reflected into the main text, keeping some of the claims vague. Although I still think this manuscript has a good potential to be published in *eLife*, I feel that the current manuscript doesn't reach a level suitable for publication.

One of the major questions was what is the potential mechanism of cell sliding after the cell chirality is neutralized (question 1). However, this is not clearly explained/speculated in the main text.

i) The author explained in the rebuttal that "the loss of cell chirality still continued during the cell sliding", but "the dissolution of cell chirality is a mechanically quicker process than the cell sliding". It is not clear which data showed "the loss of cell chirality still continued during the cell sliding", and which section of the text explained this statement. Moreover, it is not easy to understand the claim that the loss of cell chirality is quicker process than cell sliding, but the loss of cell chirality is in progress (or still continue) during cell sliding. It would be great if the authors expand the explanations in the main text and help the reader to understand.

ii) Later, the author claimed "cell deformation occurring during the course of cell chirality loss drives the cell sliding". It is not clear which data support this statement, and what is the rationale to support this statement.

iii) In the rebuttal letter, the authors stated "to address the point raised by the reviewer, we modified our description of the loss of cell chirality as follows". However, what the authors did were 1) making some words changes including from "the cancellation of cell chirality" to "the loss of cell chirality", and 2) adding a statement "which was reminiscent of the cell sliding observed in vivo". Basically, no scientific information/explanation was added in the main text to answer the question. I would strongly suggest the authors to add words/sentences in the main text to clarify the original question: what is the potential mechanism (active driving forces) of cell sliding after the cell chirality is neutralized (and which data support the claim)?

Question 1: The author added a sentence in the Discussion section: "…the driving force for hindgut rotation resides in the hindgut epithelial cells themselves". This statement was clearly supported by the explant experiment. However, it is not clear to me how this statement supports the idea "excluding the possibility that chiral cell sliding is a consequence of the hindgut tube rotation". This could be just a wording issue. It would be great if the author expand the Discussion section and help the reader to understand. It is possible that this ambiguity could be eliminated once the mechanism of cell sliding is clearly explained.

Question 2: The author's responses were "As also pointed out in our reply to Comment 1, there is a cause and effect relationship between the dissolution of cell chirality and the cell sliding. Regarding this issue, please see our reply to Comment 1." It is still not clear where the cause-and-effect relationship came from and which data support this relationship. I do not feel that the comment 1 by the authors did not answer to the question. Please see the comments above.

---

## [Author Response]

Essential revisions:1) […] The evidence to claim that the "cell sliding" is an active process, which drives tissue rotation, is weak. The major data the authors put forward to claim this point is the in silico results showing the cancellation of cell chirality precedes the cell sliding (and tissue rotation). Based on this observation, the authors argue that the cell sliding is caused by the cancellation of cell chirality. This claim raises a few important questions.What would be the mechanisms (or driving force) of cell sliding (and tissue rotation) under the circumstance where the cell chirality is neutralized before the rotation? If the tissue rotation is a result of the viscoelastic response of the tissue after "relaxing procedure", then how this response cause only the cell sliding during tissue rotation but not the cell chirality change?

Thank you very much for this valuable comment. We realized that our description that the dissolution of cell chirality precedes the cell sliding was oversimplified and not clear enough. We observed that the loss of cell chirality began before the cells started shifting their positions, and that the loss of cell chirality still continued during the cell sliding. In other words, the dissolution of cell chirality is a mechanically quicker process than the cell sliding. However, cell deformation occurring during the course of cell chirality loss drives the cell sliding. We believe that these two phases overlap each other and cannot be clearly divided. We also note that it is technically very difficult to measure the precise time lag between cell-chirality loss and cell sliding in vivo. In the revised manuscript, to address the point raised by the reviewer, we modified our description of the loss of cell chirality as follows in the Results, subsection “Simulation of cell chirality-induced cellular behavior by a vertex model”:

“We found that in the early phases of the simulation, the polygons quickly became rounder or less polarized in shape, suggesting that the loss of cell chirality began even before the polygons started to prominently change their relative positions (Figure 4E, *t* = 1.0). Subsequently (*t* = 80.0), the polygons changed their relative positions slowly in a leftward direction (Figure 4E, *t* = 80.0), which was reminiscent of the cell sliding observedin vivo. Thus, the loss of cell chirality began before the changes in the relative positions of the cells.”

Moreover, it is not clear whether this time lag between the cancellation of cell chirality and the cell sliding can be observed in Drosophila embryonic hindgut.

We agree with this comment. We tried to develop an imaging procedure to simultaneously detect the changes in the shape of cell boundaries and the position of nuclei in vivo. However, due to the thickness of the hindgut, it was difficult to obtain high-resolution time-lapse images of the nuclei and cell boundaries for this analysis. Thus, although the experiments suggested by the reviewer could be very informative, we could not perform them.

Overall, the current manuscript is not sufficient to reject the possibility that the cell sliding observed here was a consequence of tissue rotation, i.e., a passive process. The authors need to carefully discuss these contrary possibilities.

Thank you very much for this important suggestion for improving the clarity of our manuscript. We believe that the cell sliding causes the tissue rotation, since the rotation occurs tissue-intrinsically, as shown by organ culture (Figure 1—figure supplement 1), and cell deformation was previously shown to be the primary driving force for the tissue rotation (Hatori et al., 2014). Thus, the possibility that the hindgut twisting actively induces cell sliding can be excluded. However, as pointed out by the reviewer, we should discuss this point to help readers understand the concept, so we added the following sentences to the Discussion, subsection “Cell chirality causes LR asymmetric organogenesis”:

“This causal relationship is also supported by our previous genetic and current explant experiments, which revealed that the driving force for hindgut rotation resides in the hindgut epithelial cells themselves, essentially excluding the possibility that chiral cell sliding is a consequence of the hindgut tube rotation.”

2) The term "cell sliding" can be misleading, since, for most cases, the adjacent cells still share the same boundary without moving away from each other. Cell sliding needs a clear definition. Is it the cell shape change?

The reviewer points out that our definition of cell sliding was not clear enough. To address this issue, we clarified the definition of cell sliding. As we defined it in the manuscript, cell sliding is a cellular behavior in which cells LR-directionally change their position relative to their subjacent neighbors by sliding in one direction. As the reviewer supposes, our model suggests that the cell deformation caused by the dissolution of cell chirality drives cell sliding.

Results, subsection “Hindgut epithelial cells show chiral sliding”

“These results together suggested that during the counterclockwise rotation of the hindgut, cells change their position relative to their subjacent cells (posterior neighbors) by sliding in the direction of rotation (Figure 1C, D, F). Here, we refer to this novel cellular behavior as “cell sliding.””

Even if we can exclude the possible role of cell intercalation, it is not clear how the authors reach the conclusion that it is chiral cell sliding that drives asymmetry. What drives the chiral cell sliding then? I suppose that it is planar cell shape chirality.

As also pointed out in our reply to Comment 1, there is a cause and effect relationship between the dissolution of cell chirality and the cell sliding. Regarding this issue, please see our reply to Comment 1.

How do the two differ from each other?

Our results demonstrated that the cell sliding and cell-chirality loss are mechanically different from each other in silico. First, the loss of cell chirality is a mechanically quicker process than cell sliding (Figure 4C-E). Second, if the tube rotation was prevented, the cell chirality was still lost even without cell sliding, suggesting that the loss of cell chirality does not depend on the cell sliding (Figure 4—figure supplement 3). In the revised manuscript, these results were described more clearly, as we addressed in our replies to Comments 1 and 2. Please see our responses to these comments.

3) The section on chiral properties of cells doesn't seem sufficient to induce a chiral bias. The determination of special edges needs to be biased towards a specific direction to induce asymmetry. Also, how is the deflection angle defined? Was it used as an input for the simulation?

As pointed out by the reviewers, the section on chiral properties in the supplement did not include a description of the introduction of chiral bias in the previous version of manuscript. Therefore, we wrote the following “initial tube” section, which includes all of the information requested by the reviewer. However, to make the information easier to find, in the revised manuscript, we combined sections and revised the descriptions as follows. We also changed the section order and put this section as the first section to show most essential information to grasp the concept of our model.

Subsection “Initial model tube for the computer simulations including cells with chiral properties”:

“We made an initial tube recapitulating the in vivo hindgut, in which the cell axes tilted leftward as shown in Figure 4C, as follows. […] The vertex dynamics stopped at *t* = 5.0. As a result, we obtained a tube as described in Figure 4A (*t*= 0).”

4) In the numerical simulation, the neighbor exchange is enabled. What if it is disabled? Will the asymmetric looping be still possible? What will happen to the angle of twisting?

We found the simulation suggested by the reviewer to be very valuable for improving our manuscript. We generated a simulation in which the neighbor exchange was disabled, as shown in Figure 4B. This model tube also rotated, just like the original model hindgut with junctional remodeling, supporting our hypothesis that cell rearrangement is not required for the tube rotation. To describe this result, we added the following sentences.

Results, subsection “Simulation of cell chirality-induced cellular behavior by a vertex model”:

“To confirm that cell deformation rather than cell intercalation is a major driving force of the cell sliding and hindgut rotation, we generated a computer simulation that did not allow junctional remodeling (Figure 4B). […] In conclusion, our in vivo and *in silico* analyses both demonstrated that cell sliding converts the intrinsic chirality of the cell shape into the LR asymmetric epithelial morphogenesis though chiral cell deformation.”

Materials and methods, subsection ““Elementary process of reconnecting neighboring vertices”:

“We also performed a computer simulation that did not allow the reconnection of vertices to test whether cell intercalation is required for the hindgut rotation (Figure 4B).”

5) Could cell intercalation be possibly a mechanism for amplifying the asymmetry? Will all cells tilted, and the intercalation can increase the total length without inducing too much stretching/elongation of the cells?

Thank you very much for this comment. We realized that the possibility suggested by the reviewer is very attractive, and we added following sentence to include it.

Results, subsection “Cell sliding appears to depend on cell-shape change but not junctional remodeling”:

“However, it might also amplify the LR asymmetry by increasing the length of the tilted cell columns.”

6) There are significant concerns regarding the organization of this manuscript. The Table 1 and Figure 3—video 3 may be used as a motive for the entire study, instead of merely mentioning them towards the end. It may be more reasonable to follow the table and video with experimental findings on cell sliding and end with numerical simulation.

We found this comment to be valuable for improving our manuscript. We reorganized the entire Results section so that our experimental findings were presented first, followed by the numerical simulations. We believe that the logical flow of information is improved in the revised manuscript.

[Editors' note: further revisions were requested prior to acceptance, as described below.]

Reviewer #1:

This manuscript studied cellular biophysics involving in the left-right asymmetric twisting of Drosophila embryonic hindgut. They developed a 3D tubular model for studying gut asymmetric rotation, and proposed that cell asymmetric sliding instead of cell intercalation determines the directionality of gut looping. Overall, the research topic is important and interesting. With the revision, the quality of manuscript has largely improved. However, there are still some concerns for the authors to consider:1) One confusing part of the manuscript is regarding the relationships between cell deformation, cell sliding, cell chirality (and its loss) and tissue rotation. This is also reflected in previous comments raised by the reviewers and editor. It is partially due to the adopted definition of cell chirality in the manuscript, which is defined as the directional biases of major axes, or cell alignment (Figure 3). However, based on this definition, cell alignment direction geometrically changes with individual cell deformation, which drives cell sliding at a multicellular level. Therefore, all these three are coupled with one another, and, in my opinion, that based on the current definition of "observed cell chirality", it is impossible to distinguish between them or tell which ones occur first and later.

Thank you very much for this comment, which is valuable for improving the clarity of our manuscript. The reviewer points out that the “cell alignment direction geometrically changes with individual cell deformation.” However, we believe that this is not the case, as shown in our computer simulations. First, as shown in Figure 4C, D, the cancellation of cell chirality occurs more quickly than the cell sliding, because the period when cell chirality was actively cancelled (t = 0 to 1.0) preceded the onset of cell sliding (after t = 1.0). Thus, in this initial period, roughly speaking, only cell deformation and not the geometrical change in cell alignment (designated as cell sliding) occurs. This observation indicates the order of the events: the cell deformation occurs first and then the cell sliding is induced. We speculate that the temporal difference in the onsets of cell deformation versus cell sliding may reflect the distinct mechanical properties of these two events, such as their viscoelasticity (please see the response to iii of reviewer #3). Such a time lag between the deformation of individual cells and tissue was also previously investigated in other vertex models (Honda et al., 2004). Second, as shown in Figure 4—figure supplement 3, the cancellation of cell chirality was observed without cell sliding, if the rotation of the model gut tube was prohibited. Therefore, the cancellation of cell chirality is not always coupled with cell sliding, demonstrating that they are two different steps with distinct mechanical properties.

Based on these results, we concluded that the cell sliding is preceded by the cancelation of cell chirality. However, we realize that our description of the above idea was not clear enough in the text of the previous manuscript, although we had discussed it in response to the reviewer’s comments. To describe this idea clearly, we added the following sentences to the revised manuscript. We also modified some sentences to describe the hypothesis more clearly.

Results, subsection “Simulation of cell chirality-induced cellular behavior by a vertex model”:

“This rapid cell deformation was also observed quantitatively, as the initial leftward biases of the cell axes (dark-colored bars, Figure 4C) and cell boundaries (dark-colored bars, Figure 4D) were largely abolished at *t* = 1.0 (light-colored bars, Figure 4C,D). […] The time lag between the initiations of cell deformation and of cell sliding suggests that these two events were mechanically distinct, probably due to the different viscoelastic properties of individual cells versus cell aggregates in our model (Honda et al., 2004).”

Same section, added:

“These results also showed that the loss of cell chirality and the cell sliding are not always coupled, further suggesting that they are distinct mechanical events.”

Discussion, subsection “Cell chirality causes LR asymmetric organogenesis”:

“In our computer simulation, we recapitulated the LR directional axial rotation of the gut tube, which is driven by a mechanical force bias induced by cell chirality. […] In turn, completion of the cell sliding led to global stabilization of the model tube.

Reviewer #3:

Overall, the author responded to the reviewers’ comments nicely in the rebuttal. However, I personally feel that many of the comments were not reflected into the main text, keeping some of the claims vague. Although I still think this manuscript has a good potential to be published in eLife, I feel that the current manuscript doesn't reach a level suitable for publication.One of the major questions was what is the potential mechanism of cell sliding after the cell chirality is neutralized (question 1). However, this is not clearly explained/speculated in the main text.i) The author explained in the rebuttal that "the loss of cell chirality still continued during the cell sliding", but "the dissolution of cell chirality is a mechanically quicker process than the cell sliding". It is not clear which data showed "the loss of cell chirality still continued during the cell sliding", and which section of the text explained this statement. Moreover, it is not easy to understand the claim that the loss of cell chirality is quicker process than cell sliding, but the loss of cell chirality is in progress (or still continue) during cell sliding. It would be great if the authors expand the explanations in the main text and help the reader to understand.

It was our omission not to refer to the results demonstrating that the loss of cell chirality still continued during the cell sliding. In this study, using the improved model hindgut, our findings suggested that the cancellation of cell chirality and the cell sliding are distinct mechanical events. Importantly, the cancellation of cell chirality (also called cell deformation in the manuscript) occurs more quickly (mostly from t = 0 to t = 1.0) than the cell sliding (after t = 1.0) as shown in Figure 4C, D (please compare the dark-colored bars with the light-colored ones). However, in Figure 4C, D, the cancellation of cell chirality based on two criteria, the LR-biased polygon axes and edges, still continued from t = 1.0 to t = 80.0 (please compare the light-colored bars with the white ones), although the degrees of the cancelation were less prominent compared with those at t = 0 to 1.0. On the other hand, as shown in Figure 4A, E, the cell sliding started after t = 1.0. Based on these observations, we concluded that the loss of cell chirality is a quicker process than the cell sliding, but the loss of cell chirality still slowly continues during the cell sliding. However, as pointed out by the reviewer, these interpretations and discussions were insufficient or were missing from the manuscript. Thus, to address these important points, we added the following sentences to the revised manuscript.

Results, subsection “Simulation of cell chirality-induced cellular behavior by a vertex model”:

“This rapid cell deformation was also observed quantitatively, as the initial leftward biases of the cell axes (dark-colored bars, Figure 4C) and cell boundaries (dark-colored bars, Figure 4D) were largely abolished at *t* = 1.0 (light-colored bars, Figure 4C, D).”

“During cell sliding, the cell-shape change became less prominent but still continued, given that the LR biases observed at *t* = 1.0 were further randomized at *t* = 80.0 (compare light-colored and white bars, Figure 4C, D). […] The time lag between the initiations of cell deformation and of cell sliding suggests that these two events were mechanically distinct, probably due to the different viscoelastic properties of individual cells versus cell aggregates in our model (Honda et al., 2004).”

ii) Later, the author claimed "cell deformation occurring during the course of cell chirality loss drives the cell sliding". It is not clear which data support this statement, and what is the rationale to support this statement.

The sentence was incorrect, and there was confusion in our usage of “cell chirality loss.” Instead, we should state that the “mechanical force biases induced by cell chirality drive the cell sliding.” To correct this sentence, we replaced previous sentences with revised ones as follows. For more details, please see the response to question 2 from reviewer #3.

Results, subsection “Simulation of cell chirality-induced cellular behavior by a vertex model”:

“Therefore, this simulation further supported the idea that the mechanical force bias induced by cell chirality drives directional cell sliding and gut-tube rotation.”

iii) In the rebuttal letter, the authors stated "to address the point raised by the reviewer, we modified our description of the loss of cell chirality as follows". However, what the authors did were 1) making some words changes including from "the cancellation of cell chirality" to "the loss of cell chirality", and 2) adding a statement "which was reminiscent of the cell sliding observed in vivo". Basically, no scientific information/explanation was added in the main text to answer the question. I would strongly suggest the authors to add words/sentences in the main text to clarify the original question: what is the potential mechanism (active driving forces) of cell sliding after the cell chirality is neutralized (and which data support the claim)?

This was also our omission not to describe these mechanisms in the previous manuscript. We believe that the mechanical force biases induced by cell chirality drive the cell sliding. In our computer simulations, the anisotropic (left-right asymmetric) contraction force that was introduced as the initial condition was equalized (contraction force becomes isotropic), which induced subsequent events, because it was the only mechanical perturbation introduced into this simulation. Under the neutralization process of the LR-biased mechanical forces, cell boundaries with less contraction force in the initial condition contracted more strongly than before, and vice versa. This balancing process was detected as the loss of cell-shape chirality mostly from t = 0 to t = 1.0 in Figure 4C, D (please compare the dark-colored bars with the light-colored ones). However, the active period for cell-chirality loss did not coincide with the initiation of the cell sliding. Our analysis revealed that the cell sliding was initiated after t = 1.0, which was preceded by the cell-chirality loss (Figure 4A, E). We speculate that the mechanical force had not yet reached equilibrium at this time (t = 1.0 and after), as the cell-chirality loss still continued, although to a lesser extent, from t = 1.0 to t = 80.0 (please compare the light-colored bars with the white ones in Figure 4C, D). Due to the viscoelastic nature of the computer-model tissue, the mechanical force biases induced by cell chirality probably did not cause the cell sliding as quickly as the cell-chirality loss in shape. The time lag between the deformation of individual cells and of tissue in other vertex models was previously reported and analyzed (Honda et al., 2004, Honda et al., 2008). Therefore, we speculate that such an unbalanced force existing at t = 1.0 and after was the ultimate driving force of the cell sliding.

However, as pointed out by the reviewer, we realize that we did not discuss these potential mechanisms in the previous manuscript. Therefore, to address this point, we added the following sentences to our revised manuscript.

Results, subsection “Simulation of cell chirality-induced cellular behavior by a vertex model”:

“During cell sliding, the cell-shape change became less prominent but still continued, given that the LR biases observed at *t* = 1.0 were further randomized at *t* = 80.0 (compare light-colored and white bars, Figure 4C, D). […] The time lag between the initiations of cell deformation and of cell sliding suggests that these two events were mechanically distinct, probably due to the different viscoelastic properties of individual cells versus cell aggregates in our model (Honda et al., 2004).”

Subsection “Cell chirality causes LR asymmetric organogenesis”:

“In our computer simulation, we recapitulated the LR directional axial rotation of the gut tube, which is driven by a mechanical force bias induced by cell chirality. […] In turn, completion of the cell sliding led to global stabilization of the model tube.”

Question 1: The author added a sentence in the Discussion section: "…the driving force for hindgut rotation resides in the hindgut epithelial cells themselves". This statement was clearly supported by the explant experiment. However, it is not clear to me how this statement supports the idea "excluding the possibility that chiral cell sliding is a consequence of the hindgut tube rotation". This could be just a wording issue. It would be great if the author expand the Discussion section and help the reader to understand. It is possible that this ambiguity could be eliminated once the mechanism of cell sliding is clearly explained.

Regarding the causal relationship, we thought that cell sliding could be passively induced in vivo if the gut tube was rotated by an external force. However, our organ culture experiment showed that the hindgut by itself, and not the surrounding tissues and organs, generates the force for driving its rotation. This excludes the possibility that the hindgut rotation is a cause of cell sliding. However, we realized that our explanation of this assumption was not clear enough. To address this issue, we modified the sentence as below.

Discussion, subsection “Cell chirality causes LR asymmetric organogenesis”:

“Our conclusions about the mechanical properties of chiral cell sliding were based on the computer simulation. […] This result also excludes the possibility that the cell sliding is a consequence of hindgut rotation driven by an external force, further supporting the validity of our computer simulation for analyzing the relationship between cell chirality loss and cell sliding.”

Question 2: The author's responses were "As also pointed out in our reply to Comment 1, there is a cause and effect relationship between the dissolution of cell chirality and the cell sliding. Regarding this issue, please see our reply to Comment 1." It is still not clear where the cause-and-effect relationship came from and which data support this relationship. I do not feel that the comment 1 by the authors did not answer to the question. Please see the comments above.

The confusion might have come from our incorrect usage of “dissolution of cell chirality,” “loss of cell chirality,” and “cell chirality loss,”. In our computer simulations, the anisotropic (left-right asymmetric) contraction force that was introduced as the initial condition was equalized (contraction force becomes isotropic), which induced all of the subsequent events, because it was the only mechanical perturbation introduced into this simulation. Thus, the “dissolution of cell chirality” in the forces was a cause of cell sliding. However, we carelessly used these three wordings to describe two different conditions: 1) the equalization of LR-biased mechanical force balance (cell chirality in force), and 2) the loss of cell chirality in cell shape. Therefore, in several sentences, our statements could be interpreted as “loss of cell chirality in cell shape drives the cell sliding,” although these sentences should have stated that the “dissolution of cell chirality in force balance” drives the cell sliding. As pointed out by the reviewer, we agree that the cause and effect relationship between the “dissolution of cell chirality in cell shape” and the cell sliding was not shown; instead, we showed the potential difference and order of these events.

To solve this problem, we replaced parts of the problematic sentences in several parts of the revised manuscript, as suggested by reviewer #1. We believe that the logic flows more smoothly in the revised manuscript. The corrected sentences are listed below.

Results, subsection “Simulation of cell chirality-induced cellular behavior by a vertex model”:

“Therefore, this simulation further supported the idea that the mechanical force bias induced by cell chirality drives directional cell sliding and gut-tube rotation.”

“Thus, collectively, the simulations indicated that cell sliding induced by a cell-chirality-provoked mechanical force bias is a dynamic cellular process connecting cell chirality and LR asymmetric tube twisting.”

Discussion, subsection “Cell chirality causes LR asymmetric organogenesis”, added:

“In our computer simulation, we recapitulated the LR directional axial rotation of the gut tube, which is driven by a mechanical force bias induced by cell chirality.”